# Rethinking Open-set Noise in Learning with Noisy Labels

## Abstract

To reduce reliance on labelled data, learning with noisy labels (LNL) has gained increasing attention. However, prevailing works typically assume that such datasets are primarily affected by closed-set noise (where the true/clean labels of noisy samples come from another known category), and ignore therefore the ubiquitous presence of open-set noise (where the true/clean labels of noisy samples may not belong to any known category). In this paper, we formally refine the LNL problem setting considering the presence of open-set noise. We theoretically analyze and compare the effects of open-set noise and closed-set noise, as well as the effects between different open-set noise modes. We also analyze common open-set noise detection mechanisms based on prediction entropy values. To empirically validate the theoretical results, we construct two open-set noisy datasets - CIFAR100-O/ImageNet-O and introduce a novel open-set test set for the widely used WebVision benchmark. Our work suggests that open-set noise exhibits qualitatively and quantitatively distinct characteristics, and how to fairly and comprehensively evaluate models in this condition requires more exploration.

## 1 Introduction

In recent years, the tremendous success of machine learning often relies on the assumption that data labels are accurate and free from noise. However, in real-world scenarios, label noise caused by factors such as annotation errors and label ambiguity is ubiquitous, posing a pervasive challenge to the performance and generalization of models. To address this challenge, various methods have been proposed to learn with noisy labels, including noise transition matrix [7, 23], label correction [17, 3], robust loss functions [6, 29, 19], and recently dominant sample selection-based approaches [11, 2].

Most current efforts, however, primarily focus on closed-set noise, where the true labels of noisy samples belong to another known class. This includes common noise models like symmetric noise (assuming that the labels of samples are randomly flipped with a certain probability to any other known classes) or asymmetric noise model (assuming that the probability of label confusion is influenced by the classes, such as 'cat' being more likely to be confused with 'dog' than with 'airplane'). Recent advancements have also explored instance-dependent noise models [4, 26], where label confusion depends directly on individual instances.

Unfortunately, unlike the in-depth exploration of closed-set noise, there is noticeably limited research on open-set noise, where the true labels of noisy samples may not belong to any known category. This gap becomes particularly crucial when considering one of the primary motivations for learning with noisy labels: learning with datasets obtained through web crawling. Examining one of the most commonly used benchmarks - the WebVision dataset [12], we validate the prevalence of open-set noise (fig. 1). In fact, the 'open-world' assumption involving open-set samples has received more attention in other weakly supervised learning problems, such as open-set recognition and outlier

Figure 1: Example images of class "Tench" from WebVision dataset. Clean samples are marked in êxtcolorgreenGreen, closed-set noise is marked in Blue and open-set noise is marked in Red. See appendix F for more details.

detection, but lacks enough exploration in the context of LNL. To this end, we focus on a thorough theoretical analysis of open-set noise in this paper. Specifically:

- Considering the presence of open-set noise, we introduce the concept of a complete noise transition matrix and reformulate the LNL problem and label noise definition in this context.
- To enable offline analysis, we consider two pragmatic cases: *fitted case*, that the model perfectly fits the noisy distribution, and *memorized case*, that the model completely memorises the noisy labels.
- We analyze and compare the open-set noise *vs.* closed-set noise on closed-set classification accuracy and suggest that open-set noise has a less negative impact in both cases. We also analyze and compare the 'hard' open-set noise *vs.* 'easy' open-set noise, but find that these two different noise modes show opposite trends in two different cases.
- Since closed-set classification evaluation may be insufficient to fully reflect model performance, we consider introducing an additional open-set detection task and conduct preliminary experiments.
- We derive and analyze the open-set noise detection mechanism based on the entropy values of model predictions and suggest that it may be effective only for 'easy' open-set noise. We also consider two representative LNL methods and combine them with such open-set noise detection mechanism for further experiments.
- For controlled experiments, we construct two novel synthetic open-set noise datasets: CIFAR100-O and ImageNet-O. Additionally, we introduce a new open-set test set to the WebVision dataset for the open-set detection task.

## 2   Related works

Methods for learning with noisy labels can be roughly categorized into two main directions. The first direction typically focuses on estimating noise transition matrix [4, 26, 23, 7] or designing robust loss functions [29, 19, 6], aiming to achieve theoretically risk-consistent or probabilistic-consistent models. However, most of these works often assume an ideal scenario where the model can learn to fit the sampled distribution well, overlooking the over-fitting issues arising from excessive model capacity and insufficient data in practical situations. *In this paper, we introduce the concept of complete noise transition matrix considering the presence of open-set noise and conduct theoretical analyses and experimental validations for both ideal case and over-fitting case, namely **fitted case and memorized case**.* The second type is often based on sample selection strategies, involving also different regularization terms and off-the-shelf techniques such as semi-supervised learning and model co-training, to achieve the state-of-the-art performance. Most sample selection methods are based on the model's current predictions, such as the popular 'small loss' mechanism [2, 11, 8, 28, 10, 17, 13, 27, 24, 30], or model's feature space [21, 22, 15, 5].

Especially, the investigation on open-set noise is relatively scarce. Wang et al. [18] utilize Local Outlier Factor algorithm to identify open-set noise in feature space, Wu et al. [22] propose to identify open-set noise with subgraph connectivity, while both Sachdeva et al. [16] and Albert et al. [1] try to identify open-set noise based on entropy-related dynamics. Instead, Feng et al. [5] do not identify open-set noise explicitly while avoid relabelling and including open-set noise in the training. More closely related to our work, Xia et al. [25] also investigates noise transition matrices involving open-set noise but considering all open-set noise belonging to a single meta-class. In this paper, we consider

that open-set noise may originate from different classes, and based on this premise, we analyze two distinct open-set noise modes. Wei et al. [20] propose leveraging open-set noise to mitigate the impact of closed-set noise, as it helps alleviating the model's over-fitting tendency. Instead, we focus on a thorough theoretical analysis of the effects with different noise modes, including open-set noise versus closed-set noise, and different open-set noise versus each other.

## 3 Methodology

In section 3.1, we briefly introduce the traditional problem formulation of LNL. In section 3.2, we reformulate the LNL problem considering open-set noise. In section 3.3, we formalize how label noise influences model generalization, particularly, on the proposed error rate inflation metric. In section 3.4, we analyze and compare the impact of open-set *vs.* closed-set noise, as well as 'easy' open-set noise *vs.* 'hard' open-set noise. In section 3.5, we scrutinize the open-set noise detection mechanism based on model prediction entropy values.

### 3.1 Traditional formulation of LNL

Supervised classification learning typically assumes that we sample a certain number of independently and identically distributed training samples $\{\boldsymbol{x}_k, y_k\}_{k=1}^{K}$ from a joint distribution $P(\mathbf{x}, \mathbf{y}; \mathbf{y} \in \mathcal{Y}^{in})$, i.e., the so-called train set. By default, here all the possible values for $y_k$ in the discrete label space $\mathcal{Y}^{in} : \{1, 2, ..., A\}$ (referred here as *inlier classes*), are known in advance. With a certain loss function, given the train set $\{\boldsymbol{x}_k, y_k\}_{k=1}^{K}$ we aim to train a model $f : \boldsymbol{x} \to y$ whose predictions can achieve the minimum error rate under the whole clean distribution $P(\mathbf{x}, \mathbf{y}; \mathbf{y} \in \mathcal{Y}^{in})$.

Under LNL problem setting, we believe that the joint distribution $P(\mathbf{x}, \mathbf{y}; \mathbf{y} \in \mathcal{Y}^{in})$ has been perturbed to $P^n(\mathbf{x}, \mathbf{y}; \mathbf{y} \in \mathcal{Y}^{in})$; especially, the conditional distribution $P^n(\mathbf{y}|\mathbf{x}; \mathbf{y} \in \mathcal{Y}^{in})$ changes — normally we assume the sampling prior is free of the label noise ($P(\mathbf{x}; \mathbf{y} \in \mathcal{Y}^{in}) = P^n(\mathbf{x}; \mathbf{y} \in \mathcal{Y}^{in})$), leading to the presence of noisy labels $y_k^n$ in the noisy train set $\{\boldsymbol{x}_k, y_k^n\}_{k=1}^{K}$ that do not conform to the clean conditional distribution $P(\mathbf{y}|\mathbf{x}; \mathbf{y} \in \mathcal{Y}^{in})$.

### 3.2 Revisiting LNL considering open-set noise

We here formally revisit the problem formulation of learning with noisy labels considering the existence of open-set noise. Instead of assuming all the possible classes are known ($\mathbf{y} \in \mathcal{Y}^{in}$), we consider samples from some unknown outlier classes may also exist in the train set. Let us denote these classes as *outlier classes* $\mathcal{Y}^{out} : \{A + 1, A + 2, ..., A + B\}$ with $B$ as the number of possible outlier classes. Then, we expand the support of joint distribution to contain both inlier and outlier classes, denoted as $P(\mathbf{x}, \mathbf{y}; \mathbf{y} \in \mathcal{Y}^{in} \cup \mathcal{Y}^{out})$ and $P^n(\mathbf{x}, \mathbf{y}; \mathbf{y} \in \mathcal{Y}^{in} \cup \mathcal{Y}^{out})$ for the clean and noisy ones, respectively. For brevity, we denote as $\mathcal{Y}^{all} \triangleq \mathcal{Y}^{in} \cup \mathcal{Y}^{out}$. Similarly as above, we still assume the noisy labelling will not affect the sampling prior ($P(\mathbf{x}; \mathbf{y} \in \mathcal{Y}^{all}) = P^n(\mathbf{x}; \mathbf{y} \in \mathcal{Y}^{all})$). For subsequent analysis, we first define below complete noise transition matrix:

**Definition 3.1** (Complete noise transition matrix). For a specific sample $\boldsymbol{x}$, we define as $T$ (sample index omitted here for simplicity) the complete noise transition matrix[1]:

$$T = \left[ \begin{array}{c|c} T^{in}_{\text{A} \times \text{A}} & \mathbf{0}_{\text{A} \times \text{B}} \\ \hline T^{out}_{\text{B} \times \text{A}} & \mathbf{0}_{\text{B} \times \text{B}} \end{array} \right].$$

$T^{in}$ corresponds to the confusion process between inlier classes $\mathcal{Y}^{in} : \{1, 2, ..., A\}$, and $T^{out}$ corresponds to the confusion process from outlier classes $\mathcal{Y}^{out} : \{A + 1, A + 2, ..., A + B\}$ to inlier classes $\mathcal{Y}^{in} : \{1, 2, ..., A\}$.

For brevity, we denote as $T_{ij} \triangleq P(\mathbf{y}^n = j | \mathbf{y} = i, \mathbf{x} = \boldsymbol{x}; \mathbf{y}^n, \mathbf{y} \in \mathcal{Y}^{all})$. We have further $\sum_{j=1}^{A+B} T_{ij} = 1$ for $i \in \{1, ..., A + B\}$ - noise transition from each clean class sums to 1 over all possible noisy classes. With such a complete noise transition matrix $T$, we can connect the clean

---

[1]The right part of the transition matrix is all-zero as we assume in the noisy labelling process all outlier classes are confused into inlier classes, i.e., all of its samples been labelled as one of the inlier classes.

conditional distribution $P(\mathrm{y}|\mathbf{x} = \boldsymbol{x}; \mathrm{y} \in \mathcal{Y}^{all})$ with the noisy conditional distribution $P^n(\mathrm{y}|\mathbf{x} = \boldsymbol{x}; \mathrm{y} \in \mathcal{Y}^{all})$ as below:

$$P^n(\mathrm{y} = j|\mathbf{x} = \boldsymbol{x}; \mathrm{y} \in \mathcal{Y}^{all}) = \sum_{l=1}^{A+B} P(\mathrm{y} = l|\mathbf{x} = \boldsymbol{x}; \mathrm{y} \in \mathcal{Y}^{all}) \cdot T_{lj} \tag{1}$$

**Label noise** Recent works usually discriminate label noise into closed-set noise and open-set noise. Before continuing with the further discussion, we feel it is necessary to elucidate these two concepts here clearly to avoid any ambiguities, as we will try to comparably discriminate and analyze them later. Specifically, most of recent works define open-set noise as 'a sample with its true label from unknown classes but mislabelled with a known label'. Formally, we have:

**Definition 3.2** (Label noise). For sample $\boldsymbol{x}$ with clean label $y$ and noisy label $y^n$:

- When $y = y^n$, $(\boldsymbol{x}, y, y^n)$ is a clean sample;

- When $y \neq y^n$ and $y \in \mathcal{Y}^{in}$, $(\boldsymbol{x}, y, y^n)$ is a closed-set noise;

- When $y \neq y^n$ and $y \in \mathcal{Y}^{out}$, $(\boldsymbol{x}, y, y^n)$ is an open-set noise.

Specifically, we have $y \sim P(\mathrm{y} = y|\mathbf{x} = \boldsymbol{x}; \mathrm{y} \in \mathcal{Y}^{all})$ while $y^n \sim P^n(\mathrm{y} = y^n|\mathbf{x} = \boldsymbol{x}; \mathrm{y} \in \mathcal{Y}^{all})$.

However, we can only identify label noise type with $(\boldsymbol{x}, y, y^n)$ — $y, y^n$ yet to be sampled even with known conditional probability. To enable sample-wise analysis on the impact of different label noise, we further introduce below $(O_{\boldsymbol{x}}, C_{\boldsymbol{x}})$ *label noise*:

**Definition 3.3** ($(O_{\boldsymbol{x}}, C_{\boldsymbol{x}})$ label noise). For sample $\boldsymbol{x}$ with clean conditional probability $P(\mathrm{y}|\mathbf{x} = \boldsymbol{x}; \mathrm{y} \in \mathcal{Y}^{all})$ and complete noise transition matrix $T$:

$$O_{\boldsymbol{x}} = \sum_{i=A+1}^{A+B} \sum_{j=1}^{A} T_{ij} P(\mathrm{y} = i|\mathbf{x} = \boldsymbol{x}; \mathrm{y} \in \mathcal{Y}^{all}) = \sum_{i=A+1}^{A+B} P(\mathrm{y} = i|\mathbf{x} = \boldsymbol{x}; \mathrm{y} \in \mathcal{Y}^{all}),$$
$$C_{\boldsymbol{x}} = \sum_{i=1}^{A} \sum_{j=1, j \neq i}^{A} T_{ij} P(\mathrm{y} = i|\mathbf{x} = \boldsymbol{x}; \mathrm{y} \in \mathcal{Y}^{all}). \tag{2}$$

Here, $O_{\boldsymbol{x}}$ is the expected open-set noise ratio, $C_{\boldsymbol{x}}$ is the expected closed-set noise ratio. We then define sample $\boldsymbol{x}$ as an $(O_{\boldsymbol{x}}, C_{\boldsymbol{x}})$ label noise. Intuitively speaking, sample $\boldsymbol{x}$ is expected to be an open-set noise with probability as $O_{\boldsymbol{x}}$ and to be a closed-set noise with probability $C_{\boldsymbol{x}}$.

With Definition 3.3, we formalize the concept of noise ratio for the whole distribution, as the accumulated $(O_{\boldsymbol{x}}, C_{\boldsymbol{x}})$ label noise at all sample points $\boldsymbol{x} \in \mathcal{X}$:

$$N = \int_{\boldsymbol{x}} (O_{\boldsymbol{x}} + C_{\boldsymbol{x}}) \cdot P(\mathbf{x} = \boldsymbol{x}; \mathrm{y} \in \mathcal{Y}^{all}) d\boldsymbol{x} \tag{3}$$

### 3.3 Analyzing classification error rate inflation in LNL

In this section, we try to analyze the impact of different label noise. Please note, while the reformulated LNL setting encompasses outlier classes $\mathcal{Y}^{out}$, in both the training and evaluation stage, they are unknown (agnostic); the learned model $f$ is still tailored for the classification of inlier classes $\mathcal{Y}^{in}$. That is to say, the default classification evaluation protocol is still concerned with the classification error rate over the inlier conditional probability, denoted as $P^f(\mathrm{y}|\mathbf{x} = \boldsymbol{x}; \mathrm{y} \in \mathcal{Y}^{in})$.

**Error rate inflation** With $P^f(\mathrm{y}|\mathbf{x} = \boldsymbol{x}; \mathrm{y} \in \mathcal{Y}^{in})$, in the evaluation phase, for specific sample $\boldsymbol{x}$ we have its prediction as: $y^f = \arg\max_k P^f(\mathrm{y} = k|\mathbf{x} = \boldsymbol{x}; \mathrm{y} \in \mathcal{Y}^{in}) \in \mathcal{Y}^{in}$, and the corresponding expected classification error rate as:

$$E_{\boldsymbol{x}} = \sum_{\mathrm{y} \neq y_f} P(\mathbf{x}, \mathrm{y}; \mathrm{y} \in \mathcal{Y}^{in}) = (1 - P(\mathrm{y} = y_f)|\mathbf{x}; \mathrm{y} \in \mathcal{Y}^{in})) \cdot P(\mathbf{x}; \mathrm{y} \in \mathcal{Y}^{in}). \tag{4}$$

Specifically, we have the Bayes error rate corresponds to the Bayes optimal model $f^*$:

$$E_{\boldsymbol{x}}^* = (1 - \max_k P(\mathrm{y} = k|\mathbf{x} = \boldsymbol{x}; \mathrm{y} \in \mathcal{Y}^{in})) \cdot P(\mathbf{x} = \boldsymbol{x}; \mathrm{y} \in \mathcal{Y}^{in}). \tag{5}$$

To measure the negative impacts of noisy labels, we care about how much extra errors have been introduced, measured by the *error rate inflation* of learned model $f$ compared to the Bayes optimal model $f^*$:

**Definition 3.4** (Error rate inflation). With $E_{\boldsymbol{x}}^*$ as the Bayes error rate, we define the *error rate inflation* for sample $\boldsymbol{x}$ as: $\Delta E_{\boldsymbol{x}} = E_{\boldsymbol{x}} - E_{\boldsymbol{x}}^*$.

**Two pragmatic cases**   However, $P^f(\mathrm{y}|\mathbf{x} = \boldsymbol{x}; \mathrm{y} \in \mathcal{Y}^{in})$, as the prediction of the final learned model $f$, is affected by many factors (model capacity/dataset size/training hyperparameters such as training epochs, etc.), which is non-trivial to determine its specific value for an offline analysis[2]. Thus, we consider two specific pragmatic cases:

- *Fitted case*: the model perfectly fits the noisy distribution: $P^f(\mathrm{y}|\mathbf{x} = \boldsymbol{x}; \mathrm{y} \in \mathcal{Y}^{in}) = P^n(\mathrm{y}|\mathbf{x} = \boldsymbol{x}; \mathrm{y} \in \mathcal{Y}^{in})$;
- *Memorized case*: the model completely memorises the noisy labels: $P^f(\mathrm{y}|\mathbf{x} = \boldsymbol{x}; \mathrm{y} \in \mathcal{Y}^{in}) = P^{y^n}(\mathrm{y}|\mathbf{x} = \boldsymbol{x}; \mathrm{y} \in \mathcal{Y}^{in})$; Here $P^{y^n}$ denotes the one-hot encoding of the noisy label $y^n$.

Nonetheless, these two cases are very realistic and important; Empirically, it is highly possible that the *memorized case* can correspond to scenarios such as scratch training based on a single-label dataset with a normal deep neural network - as normally such model has enough capacity to memorize all the labels, while the *fitted case* can correspond to scenarios such as fine-tuning a linear classifier with a pre-trained model - as the pre-trained model already captures good sample representations and the capacity of a linear classifier is limited.

## 3.4   Error rate inflation analysis *w.r.t* different label noise

In this section, we focus on analyzing the error rate inflation of different label noise. Let us recall the clean conditional distribution as $P(\mathrm{y}|\mathbf{x}; \mathrm{y} \in \mathcal{Y}^{all})$. For ease of analysis, we contemplate a simple scenario, wherein the entire clean conditional distribution remains unchanged, except only one of the sample points, say $\boldsymbol{x}$, is afflicted by label noise:

$$P^n(\mathrm{y}|\mathbf{x} \neq \boldsymbol{x}; \mathrm{y} \in \mathcal{Y}^{all}) = P(\mathrm{y}|\mathbf{x} \neq \boldsymbol{x}; \mathrm{y} \in \mathcal{Y}^{all}), \; P^n(\mathrm{y}|\mathbf{x} = \boldsymbol{x}; \mathrm{y} \in \mathcal{Y}^{all}) \neq P(\mathrm{y}|\mathbf{x} = \boldsymbol{x}; \mathrm{y} \in \mathcal{Y}^{all}). \quad (6)$$

In this condition, we can simplify analyzing the impact of label noise on the whole distribution to analyzing the error rate inflation of a single sample $\boldsymbol{x}$. Specifically, we consider two specific sample points $\boldsymbol{x}_1$ and $\boldsymbol{x}_2$, corresponding to two in our later comparative analysis. Let us denote its clean conditional probability as $P(\mathrm{y}|\mathbf{x} = \boldsymbol{x}_1; \mathrm{y} \in \mathcal{Y}^{all}) = [p_1^1, ..., p_A^1, ..., p_{A+B}^1]$ and $P(\mathrm{y}|\mathbf{x} = \boldsymbol{x}_2; \mathrm{y} \in \mathcal{Y}^{all}) = [p_1^2, ..., p_A^2, ..., p_{A+B}^2]$, and noise transition matrix as $T^1$ and $T^2$, respectively. We further assume:

$$O_{\boldsymbol{x}_1} + C_{\boldsymbol{x}_1} = O_{\boldsymbol{x}_2} + C_{\boldsymbol{x}_2} = \delta. \quad (7)$$

*We compare the error rate inflation ($\Delta E_{\boldsymbol{x}_1}$ vs $\Delta E_{\boldsymbol{x}_2}$) with different label noise given same/fixed noise ratio for a strictly fair comparison.* Note we assume that $\boldsymbol{x}_1$ and $\boldsymbol{x}_2$ hold the same sampling prior probability: $P(\mathbf{x} = \boldsymbol{x}_1; \mathrm{y} \in \mathcal{Y}^{all}) = P(\mathbf{x} = \boldsymbol{x}_2; \mathrm{y} \in \mathcal{Y}^{all})$); so that, we assure that the whole noise ratio $N$ is fixed, and more importantly, sample $\boldsymbol{x}_1$ and $\boldsymbol{x}_2$ can be considered as probabilistic exchangeable in the dataset collection process.

For better clarity, we depict the derivation relations for $\Delta_{\boldsymbol{x}}$ in fig. 2. Specifically, for our two interested cases above, we have corresponding error rate inflation for sample $\boldsymbol{x}$ (sample subscript omitted for simplicity) as:

- *Fitted case*:
$$\Delta E_{\boldsymbol{x}} = \max[p_1, ..., p_A] - p_{\arg \max[\sum_{i=1}^{A+B} p_i T_{i1}, ..., \sum_{i=1}^{A+B} p_i T_{iA}]} \quad (8)$$

- *Memorized case*:
$$\Delta E_{\boldsymbol{x}} = \max[p_1, ..., p_A] - \sum_{i=1}^{A}(p_i \cdot \sum_{j=1}^{A+B} p_j T_{ji}) \quad (9)$$

We notice that $\Delta_{\boldsymbol{x}}$ in both cases are only affected by clean conditional probability $P(\mathrm{y}|\mathbf{x} = \boldsymbol{x}_1; \mathrm{y} \in \mathcal{Y}^{all})$ and complete noise transition matrix $T$.

---

[2]The reader may refer to [14] for more discussions about related topics such as model generalization.

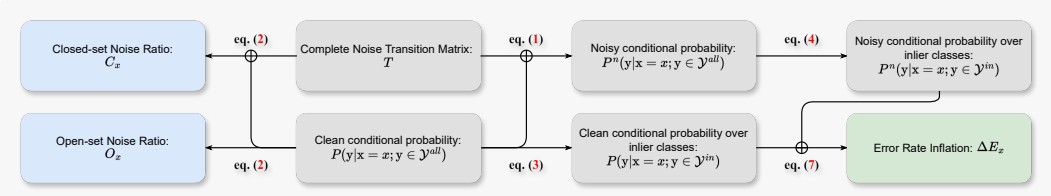

Figure 2: All-in-one derivation flowchart. Full details in appendix C.

 ### 3.4.1 How does open-set noise compare to closed-set noise?

We first try to elucidate the difference between open-set noise and closed-set noise. Without loss of generality, we consider:

$$O_{\boldsymbol{x}_1} > O_{\boldsymbol{x}_2} , \ C_{\boldsymbol{x}_1} < C_{\boldsymbol{x}_2}. \tag{10}$$

Intuitively speaking, we consider sample $\boldsymbol{x}_1$ to be more prone to open-set noise compared to sample $\boldsymbol{x}_2$, thus corresponding to the 'more open-set noise' scenario. However, without extra regularizations, there exist infinite $T^1$ and $T^2$ fulfilling eq. (7) and eq. (10) given specific $P(\mathrm{y}|\mathbf{x} = \boldsymbol{x}_1; \mathrm{y} \in \mathcal{Y}^{all})$ and $P(\mathrm{y}|\mathbf{x} = \boldsymbol{x}_2; \mathrm{y} \in \mathcal{Y}^{all})$ (see toy example below), the analysis on $\Delta E_{\boldsymbol{x}_1}$ *vs* $\Delta E_{\boldsymbol{x}_2}$ is thus infeasible.

> **Toy example about agnostic $T$** Assuming a ternary classification, with two known inlier classes ("0" and "1") and one unknown outlier class "2". Say, we have sample $\boldsymbol{x}_1$ with clean conditional probability as $[0.1, 0.2, 0.7]$. Assuming two different noise transition matrices for $T^1$ below:
>
> $$[0.55, 0.45, 0.0] = [0.1, 0.2, 0.7] \left[\begin{array}{cc|c} 0.5 & 0.5 & 0 \\ 0.75 & 0.25 & 0 \\ \hline 0.5 & 0.5 & 0 \end{array}\right]$$
>
> $$[0.45, 0.55, 0.0] = [0.1, 0.2, 0.7] \left[\begin{array}{cc|c} 0 & 1 & 0 \\ 0.5 & 0.5 & 0 \\ \hline 0.5 & 0.5 & 0 \end{array}\right]$$
>
> We have $O_{\boldsymbol{x}_1} = 0.7, C_{\boldsymbol{x}_1} = 0.2$ in both conditions but we arrive at different noisy conditional probability, similarly for sample $\boldsymbol{x}_2$.

We thus consider a class concentration assumption — in most classification datasets, the majority of samples belong to specific class exclusively with high probability. In this condition, we have proved:

**Theorem 3.5** (Open-set noise *vs* closed-set noise). *Let us consider sample $\boldsymbol{x}_1$, $\boldsymbol{x}_2$ fulfilling eq. (7) and eq. (10) - compared to $\boldsymbol{x}_2$, $\boldsymbol{x}_1$ is considered as more prone to open-set noise. Let us denote $a = \arg\max_i P(\mathrm{y} = i|\mathbf{x} = \boldsymbol{x}_1; \mathrm{y} \in \mathcal{Y}^{all})$ and $b = \arg\max_i P(\mathrm{y} = i|\mathbf{x} = \boldsymbol{x}_2; \mathrm{y} \in \mathcal{Y}^{all})$, we assume (with a high probability): $p_a^1 \to 1, \{p_i^1 \to 0\}_{i \neq a}$ and $p_b^2 \to 1, \{p_b^2 \to 0\}_{i \neq b}$. Then, we have:*

$$\Delta E_{\boldsymbol{x}_1} < \Delta E_{\boldsymbol{x}_2}$$

*in both **Fitted case** and **Memorized case**.*

Please refer to appendix D.1 for detailed proof. *To summarize, we validate that in most conditions, open-set noise is less harmful than closed-set noise in both **fitted case** and **memorized case**.*

### 3.4.2 How does different open-set noise compare to each other?

We further study how different open-set noise affect the model. Specifically, we consider:

$$O_{\boldsymbol{x}_1} = O_{\boldsymbol{x}_2} , \ C_{\boldsymbol{x}_1} = C_{\boldsymbol{x}_2} = 0. \tag{11}$$

Intuitively speaking, we focus on the impacts of different open-set noise modes given the same/fixed open-set noise ratio, while excluding the effect of closed-set noise. In this section, we assume sample $\boldsymbol{x}_1$ and sample $\boldsymbol{x}_2$ holds the same clean conditional probability: $[p_1^1, ..., p_A^1, ..., p_{A+B}^1] = [p_1^2, ..., p_A^2, ..., p_{A+B}^2]$, to only focus on the impact of different open-set noise modes with the same original sample. It is straightforward that $O_{\boldsymbol{x}_1} = O_{\boldsymbol{x}_2}$ always holds since $\sum_{i=A+1}^{A+B} p_i^1 = \sum_{i=A+1}^{A+B} p_i^2$. To ensure $C_{\boldsymbol{x}_1} = C_{\boldsymbol{x}_2} = 0$, we simply set $T_{in}^1 = T_{in}^2 = \mathbf{I}$.

Thus, we have the flexibility to explore various forms of $T_{out}$ — corresponding to different open-set noise modes. Specifically, we consider two distinct open-set noise modes: 'easy' open-set noise when the transition from outlier classes to inlier classes involves completely random flipping, and 'hard' open-set noise when there exists an exclusive transition between the outlier class and specific inlier class. We denote as $T^{easy}$ for 'easy' open-set noise and $T^{hard}$ for 'hard' open-set noise, with intuitive explanations below:

$$T^{easy} = \begin{bmatrix} \frac{1}{A} & \cdots & \frac{1}{A} \\ \cdots & \cdots & \cdots \\ \frac{1}{A} & \cdots & \frac{1}{A} \end{bmatrix}_{B \times A} \tag{12}$$

and

$$T^{hard} = \begin{bmatrix} 0 & \cdots & 1 \\ \cdots & \cdots & \cdots \\ 1 & \cdots & 0 \end{bmatrix}_{B \times A} \tag{13}$$

Especially, for $T^{easy}$, we have $T_{ij} = \frac{1}{A}$ everywhere; for $T^{hard}$, we denote as $H_i : \{\arg_j(T^{hard}_{ji} = 1)\}_{i=1}^A$ the set of corresponding outlier classes $j \in \mathcal{Y}^{out}$ confused to inlier class $i \in \mathcal{Y}^{in}$. Without loss of generality, we consider $\boldsymbol{x}_1$ with 'easy' open-set noise $T^{easy}$ and $\boldsymbol{x}_2$ with 'hard' open-set noise $T^{hard}$. Please note, that we no longer require class concentration assumption here as the noise transition matrix is already known. In this condition, we have proved:

**Theorem 3.6** ('Hard' open-set noise *vs* 'easy' open-set noise). *Let us consider sample $\boldsymbol{x}_1$, $\boldsymbol{x}_2$ fulfilling eq. (7) and eq. (11). We set the corresponding noise transition matrix as $T^1_{out} = T^{easy}, T^2_{out} = T^{hard}, T^1_{in} = T^2_{in} = \mathbf{I}$ and denote $[p^1_1, ..., p^1_A, ..., p^1_{A+B}] = [p^2_1, ..., p^2_A, ..., p^2_{A+B}] = [p_1, ..., p_A, ..., p_{A+B}]$. Then, we have:*

- *Fitted case:*

$$\Delta E_{\boldsymbol{x}_1} \leq \Delta E_{\boldsymbol{x}_2}.$$

- *Memorized case:*

$$\Delta E_{\boldsymbol{x}_1} - \Delta E_{\boldsymbol{x}_2} = \sum_{i=1}^A a_i b_i.$$

*Here, $a_i = p_i, b_i = \sum_{j \in H_i} p_j - \frac{1}{A} \sum_{i=A+1}^{A+B} p_i$.*

Please refer to appendix D.2 for detailed proof. Specifically, we further discuss about ***memorized case*** here. Since $\sum_{i=1}^A b_i = 0, \sum_{i=1}^A a_i = 1$, we can easily infer $\max(\Delta E_{\boldsymbol{x}_1} - \Delta E_{\boldsymbol{x}_2}) \geq 0, \min(\Delta E_{\boldsymbol{x}_1} - \Delta E_{\boldsymbol{x}_2}) \leq 0$. With theorem D.3, we know when the ranking of $\{p^1_i\}_{i=1}^A$ is completely in agreement with the ranking $\{\sum_{j \in H_i} p^1_j\}_{i=1}^A$ (constant term $-\frac{1}{A} \sum_{i=A+1}^{A+B} p^1_i$ omitted here), we reach its maximum value with $\Delta E_{\boldsymbol{x}_1} - \Delta E_{\boldsymbol{x}_2} \geq 0$. Intuitively speaking, this implies a scenario that the 'hard' open-set noise tends to confuse a sample into the inlier class it primarily belongs to (with higher semantic similarity), as indicated by its higher probability (the higher the $p^1_i$ the higher the $\sum_{j \in H_i} p^1_j$). For example, an outlier 'tiger' image is wrongly included as a 'cat' rather than a 'dog' in a 'cat *vs* dog' binary classification dataset. As this is more consistent with the common intuition, we default to such noise mode for 'hard' open-set noise — assuming the ranking of $\{p^1_i\}_{i=1}^A$ is of high agreement with the ranking of $\{\sum_{j \in H_i} p^1_j\}_{i=1}^A$.

*To summarize, unlike the general comparison between open-set noise and closed-set noise, the 'hard' open-set noise and the 'easy' open-set noise exhibit an opposite trend in two different cases. In the **fitted case**, 'easy' open-set noise appears to be less harmful, while in the **memorized case**, the impact of 'hard' open-set noise is comparatively smaller.*

## 3.5 Rethinking open-set noise detection

In this section, we try to investigate a commonly used open-set noise identification mechanism based on entropy dynamics. Within the sample selection paradigm, several methods [1, 16] have proposed to further identify open-set noise, based on the empirical phenomenon that samples with relatively in-confident predictions are usually open-set samples, characterized by its high prediction entropy. Specifically, we consider original sample $\boldsymbol{x}$ without noise transition, $\boldsymbol{x}$ with $T^{hard}$ and $\boldsymbol{x}$ with $T^{easy}$

as a clean sample, a 'hard' open-set noise and an 'easy' open-set noise, respectively. For simplicity, we omit the subscript.

Empirically, most sample selection method starts from the early training stages after certain epochs of warm-up training, expecting the model to learn meaningful information before over-fitting. To analyze the entropy dynamics, we thus consider the model predictions in the ***fitted case*** as a pragmatic proxy. Let us denote as $\mathcal{H}_{easy}$, $\mathcal{H}_{hard}$ and $\mathcal{H}_{clean}$ the prediction entropy corresponds to these three conditions, we have[3]:

$$
\begin{aligned}
\mathcal{H}_{clean} &= \mathcal{H}([\frac{p_1}{\sum_{i=1}^{A} p_i}, ..., \frac{p_A}{\sum_{i=1}^{A} p_i}]) \\
&= \mathcal{H}([p_1 + \frac{p_1}{\sum_{i=1}^{A} p_i} \sum_{i=A+1}^{A+B} p_i, ..., p_A + \frac{p_A}{\sum_{i=1}^{A} p_i} \sum_{i=A+1}^{A+B} p_i]), \\
\mathcal{H}_{easy} &= \mathcal{H}([p_1 + \frac{1}{A} \sum_{i=A+1}^{A+B} p_i, ..., p_A + \frac{1}{A} \sum_{i=A+1}^{A+B} p_i]), \\
\mathcal{H}_{hard} &= \mathcal{H}([p_1 + \sum_{j \in H_1} p_j, ..., p_A + \sum_{j \in H_A} p_j]).
\end{aligned}
\tag{14}
$$

We note $\mathcal{H}_{easy} \geq \mathcal{H}_{clean}$[4]. However, comparing $\mathcal{H}_{hard}$ and $\mathcal{H}_{clean}$ is non-trivial without specific values for each entry. *Thus, we suggest open-set noise detection based on the prediction entropy may only be effective for 'easy' open-set noise.*

# 4 Experiments

In this section, we try to validate our theoretical findings. In section 4.1, we validate the theoretical comparisons of different label noise. In section 4.2, we validate the entropy dynamics with different label noise. Moreover, in appendix E.1, we revisit the performance of two existing LNL methods involving open-set noise. To conduct more controllable, fair and accurate experiments, we propose two synthetic open-set noisy datasets — CIFAR100-O and ImageNet-O, respectively based on the CIFAR100 and ImageNet datasets. We also consider closed-set noise in some experiments, particularly, the symmetric closed-set noise. Please refer to appendix A for more dataset and implementation details and also details about open-set detection protocol.

## 4.1 Empirical validation on previous probabilistic findings

In this section, we conduct experiments to validate the theorem 3.5 and theorem 3.6. Since most deep models have sufficient capacity, we consider direct supervised learning from scratch on the noisy dataset and consider the final model as the ***memorized case*** - as evidenced by nearly $100\%$ train set accuracy. Conversely, obtaining a model that perfectly fits the data distribution is often challenging; here, we consider training a single-layer linear classifier upon a frozen pretrained encoder. Due to the limited capacity of the linear layer, we expect to roughly approach the ***fitted case***.

We show classification accuracy on CIFAR100-O and ImageNet-O datasets under different noise ratios, as shown in fig. 3(a/b). We find that: 1) in both cases, the presence of open-set noise has a significantly smaller impact on classification accuracy compared to closed-set noise. 2) 'hard' open-set noise and 'easy' open-set noise show opposite trends in the two different scenarios. These results align perfectly with our theoretical analysis.

In addition to closed-set classification accuracy, we also report the model's open-set detection performance using the maximum prediction value as the indicator [9]) in fig. 3(c/d). We find that, in both cases, the presence of open-set noise leads to a degraded open-set detection performance, while conversely, the presence of closed-set noise can often even enhance open-set detection performance. In light of this contrasting trend, we propose that the open-set detection task, in addition to the default closed-set classification, may help to offer a more comprehensive evaluation of LNL methods.

---

[3]Please refer to appendix D.2 for full derivation, specifically the eq. (36) and eq. (37).

[4]Please note, empirically the relative minority of open-set samples can also lead to low-confidence predictions, which is beyond the scope of this work. We leave it to interested readers.

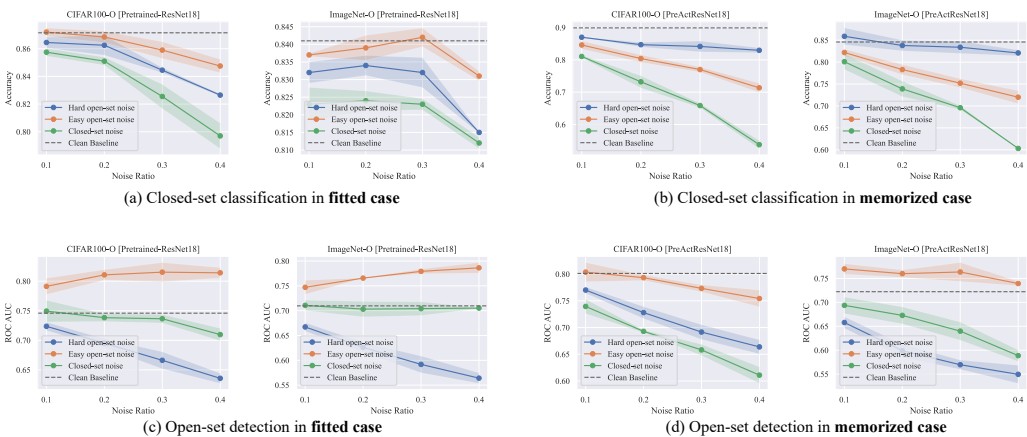

(a) Closed-set classification in **fitted case**      (b) Closed-set classification in **memorized case**

(c) Open-set detection in **fitted case**      (d) Open-set detection in **memorized case**

Figure 3: Direct supervised training with different noise modes/ratios.

## 4.2 Inspecting entropy-based open-set noise detection mechanism

In section 3.5, we briefly analyze the open-set detection mechanism based on the entropy values of model predictions and find that it may be effective only for 'easy' open-set noise. Here, we again utilize the CIFAR100-O and ImageNet-O datasets for validation experiments with different open-set noise ratios and modes. Specifically, we adopt the common warm-up idea used in existing LNL methods - training with the entire dataset for a certain number of epochs. We report the model's predicted entropy values for each sample at the $\{5th, 10th, 20th, 30th\}$ epoch in fig. 4.

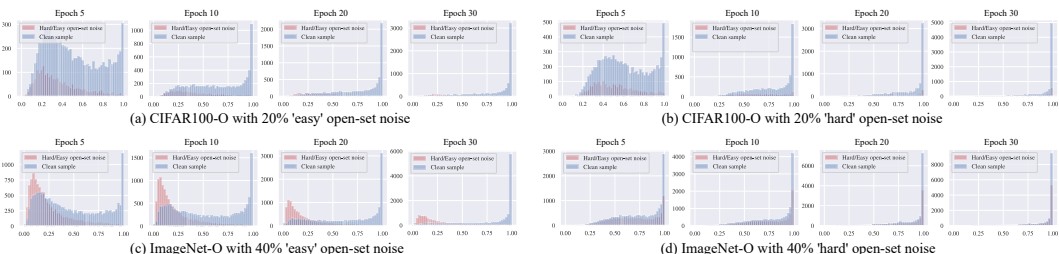

(a) CIFAR100-O with 20% 'easy' open-set noise      (b) CIFAR100-O with 20% 'hard' open-set noise

(c) ImageNet-O with 40% 'easy' open-set noise      (d) ImageNet-O with 40% 'hard' open-set noise

Figure 4: Entropy dynamics *w.r.t* different datasets/noise modes/noise ratios.

We validate that the entropy dynamics is a more effective indicator for 'easy' open-set noise compared to 'hard' open-set noise ((a) vs (b), (c) vs (d) in fig. 4). However, even for 'easy' open-set noise, we also notice that the warm-up epoch matters a lot — too early ($5th$ epoch in fig. 4(c)) or too late ($30th$ epoch in fig. 4(c)) also make open-set noise difficult to distinguish. We also test with mixed noise including both open-set noise and closed-set noise, please refer to appendix B for more discussions.

## 5 Conclusions

This paper focuses on exploring how open-set label noise affects the performance of models. While the 'open world' setting involving open-set samples has been widely discussed in several other weakly supervised learning settings, its application in the context of learning with noisy labels has been understudied. In light of this, we reconsider the LNL problem, specifically focusing on the impact of open-set noise compared to closed-set noise, and different types of open-set noise compared to each other, on the evaluation performance. In light of the challenges existing testing frameworks face in handling open-set noise, we explore the open-set detection task to address the deficiencies in model evaluation for open-set noise and conducted preliminary experiments. Additionally, we look into the common mechanism for detecting open-set noise based on the model's prediction entropy. Both theoretical and empirical results highlight the urgent need for a deeper exploration of open-set noise and its complex impact on model performance.

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

# A   Experiment details

## A.1   Dataset details

Previous works involving open-set noise also try to build synthetic noisy datasets, typically treating different datasets as open-set noise for each other to construct synthetic noisy dataset [16, 22]. In this scenario, potential domain gaps could impact a focused analysis of open-set noise. In this work, we propose selecting inlier/outlier classes from the same dataset to avoid this issue. Besides, in previous works, the consideration of open-set noise patterns often focused on random flipping from outlier classes to all possible inlier classes, which is indeed the 'easy' open-set noise adopted here. However, both our theoretical analysis and experimental findings demonstrate that 'easy' open-set noise and 'hard' open-set noise exhibit distinct characteristics. Therefore, relying solely on experiments with 'Easy' open-set noise is insufficient, emphasizing the necessity to explore and understand the complexities associated with different types of open-set noise. We also evaluate with closed-set noise in some experiments, by default, we consider the common symmetric closed-set noise in this work.

**CIFAR100-O**   For the original CIFAR100 dataset, in addition to the commonly-used 100 fine classes, there exist 20 coarse classes each consisting of 5 fine classes. To build CIFAR100-O, we select one fine class from each coarse class as an inlier class (20 classes in total) while considering the remaining classes as outlier classes (80 classes in total). Then, we consider 'Hard' and 'Easy' open-set noise as below:

- 'Hard': Randomly selected samples from the same coarse category as the target category were introduced as open-set noise.

- 'Easy': Regardless of the target category, samples from the remaining categories were randomly introduced as open-set noise.

**ImageNet-O**   For a more challenging benchmark, we consider ImageNet-1K datasets - consisting of 1,000 classes. Specifically, we randomly select 20 classes and artificially identify another 20 classes similar to each of them:

*inliers= ['tench', 'great white shark', 'cock', 'indigo bunting', 'European fire salamander', 'African crocodile', 'barn spider', 'macaw', 'rock crab', 'golden retriever', 'wood rabbit', 'gorilla', 'abaya', 'beer bottle', 'bookcase', 'cassette player', 'coffee mug', 'shopping basket', 'trifle', 'meat loaf']*

*outliers= ['goldfish', 'tiger shark', 'hen', 'robin', 'common newt', 'American alligator', 'garden spider', 'sulphur-crested cockatoo', 'king crab', 'Labrador retriever', 'Angora', 'chimpanzee', 'academic gown', 'beer glass', 'bookshop', 'CD player', 'coffeepot', 'shopping cart', 'ice cream', 'pizza']*

Then, we consider 'Hard' and 'Easy' open-set noise as below:

- 'Hard': Randomly select samples from the corresponding similar outlier class as the target category were introduced as open-set noise.

- 'Easy': Samples from the remaining categories were randomly introduced as open-set noise.

For open-set detection, we directly use the corresponding test sets of these classes from the original datasets.

**WebVision**   WebVision [12] is an extensive dataset comprising 1,000 classes of images obtained through web crawling, which thus contains a large amount of open-set noise. In line with previous studies [10, 11, 15], we evaluate our methods using the first 50 classes from the Google Subset of WebVision. To test the performance of open-set detection on the WebVision dataset, we collect a separate test set consisting of open-set images, following the same collection process as the WebVision dataset. Specifically, we utilize the Google search engine with the class names as keywords and identify those open-set samples that haven't been included in the train set for this test set.

## A.2 Implementation details

Here, we provide detailed implementation specifications for the *fitted case* and *memorized case* in section 4.1. We also briefly the applied open-set detection protocol.

**Fitted case** For the *fitted case*, we train a randomly initialized classifier - a single linear layer based on the encoder of the ResNet18 model with pretrained weights. In the case of the CIFAR100-O dataset, a weak augmentation strategy involving image padding and random cropping is applied during training, with a batch size of 512. The weight decay (wd) is set to 0.0005, and the model undergoes training for 100 epochs, utilizing a learning rate (lr) of 0.02. The learning rate schedule follows a cosine annealing strategy.

For the ImageNet-O dataset, no augmentation is applied during training. The batch size is maintained at 512, with a weight decay (wd) of 0.01. The model is trained for 100 epochs, employing a learning rate (lr) of 0.02. The learning rate schedule for this case also adheres to a cosine annealing strategy.

**Memorized case** In this case, we train a PreResNet18 model from scratch. For both datasets, a weak augmentation strategy involving image padding and random cropping is applied during training, with a batch size of 128. The weight decay (wd) is set to 0.0005, and the model undergoes training for 200 epochs, utilizing a learning rate (lr) of 0.02. The learning rate schedule also follows a cosine annealing strategy.

**Open-set detection protocol** We use the maximum softmax probability in [9] for the open-set detection task. Specifically, assume the trained model $f$ outputs a softmax vector $\boldsymbol{p}_i$ for each sample $\boldsymbol{x}_i$. We then choose a threshold value $t$ between 0 and 1. For evaluation, we consider binary labels indicating whether a sample belongs to a known class (closed-set) or the open-set and convert the open-set detection task into a binary classification problem. Samples with a maximum softmax value $p_i^{max}$ below the threshold are considered potential open-set samples. This is because a low maximum value indicates the model is less confident in any specific class for that sample.

## B Entropy dynamics for mixed label noise

In addition to the open-set noise only scenario, we also inspect the entropy dynamics with mixed label noise in fig. 5. Here, we use the notation '0.2all_0.5easy' to represent a scenario where the total noise ratio is 0.2, and within this, half of them are 'easy' open-set noise. In the presence of mixed label noise, the existence of closed-set noise severely interferes with identifying open-set noise. For example, in fig. 5(d), the entropy values of open-set noise even exceed those of clean samples. Though not theoretically analyzed, this further suggests that entropy dynamics based on model predictions, may be fragile, and we need to handle open-set noise more cautiously.

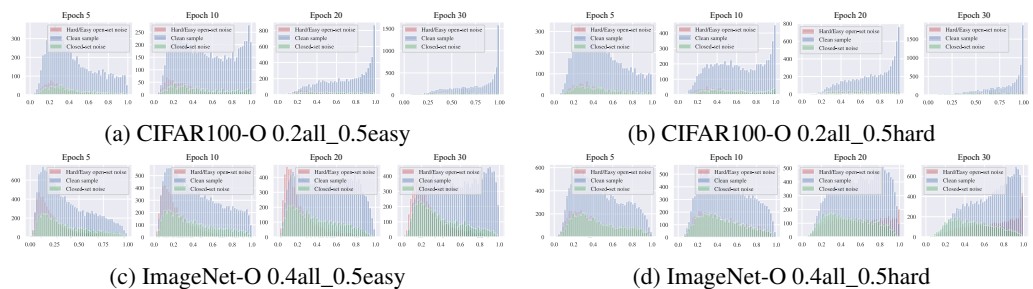

(a) CIFAR100-O 0.2all_0.5easy      (b) CIFAR100-O 0.2all_0.5hard

(c) ImageNet-O 0.4all_0.5easy      (d) ImageNet-O 0.4all_0.5hard

Figure 5: Entropy dynamics w.r.t mixed label noise.

## C Error rate inflation in two different cases

In this section, we present the computation details of error rate inflation in two interested cases - *fitted case* and *memorized case*. Specifically, we have:

- **Fitted case**:

$$E_{\boldsymbol{x}} = (1 - P(y = \arg\max_k P^n(y = k|\mathbf{x} = \boldsymbol{x}; y \in \mathcal{Y}^{in})|\mathbf{x} = \boldsymbol{x}; y \in \mathcal{Y}^{in})) \cdot P(\mathbf{x} = \boldsymbol{x}; y \in \mathcal{Y}^{in}). \tag{15}$$

- **Memorized case**:

$$\begin{aligned}
E_{\boldsymbol{x}} &= (1 - P(y = \arg\max_k P^{y^n}(y = k|\mathbf{x} = \boldsymbol{x}; y \in \mathcal{Y}^{in})|\mathbf{x} = \boldsymbol{x}; y \in \mathcal{Y}^{in})) \cdot P(\mathbf{x} = \boldsymbol{x}; y \in \mathcal{Y}^{in}) \\
&= \sum_{y^n \in \mathcal{Y}^{in}} (1 - P(y = y^n|\mathbf{x} = \boldsymbol{x}; y \in \mathcal{Y}^{in})) P^n(y = y^n|\mathbf{x} = \boldsymbol{x}; y \in \mathcal{Y}^{in}) \cdot P(\mathbf{x} = \boldsymbol{x}; y \in \mathcal{Y}^{in}) \\
&= [1 - \sum_{y^n \in \mathcal{Y}^{in}} P(y = y^n|\mathbf{x} = \boldsymbol{x}; y \in \mathcal{Y}^{in}) P^n(y = y^n|\mathbf{x} = \boldsymbol{x}; y \in \mathcal{Y}^{in})] \cdot P(\mathbf{x} = \boldsymbol{x}; y \in \mathcal{Y}^{in})
\end{aligned} \tag{16}$$

While $E_{\boldsymbol{x}}^*$ denotes the Bayes optimal error rate:

$$E_{\boldsymbol{x}}^* = (1 - \max_k P(y = k|\mathbf{x} = \boldsymbol{x}; y \in \mathcal{Y}^{in})) \cdot P(\mathbf{x} = \boldsymbol{x}; y \in \mathcal{Y}^{in}). \tag{17}$$

We thus have $\Delta E_{\boldsymbol{x}}$ in both cases as:

- **Fitted case**:

$$\begin{aligned}
\Delta E_{\boldsymbol{x}} &= [\max_k P(y = k|\mathbf{x} = \boldsymbol{x}; y \in \mathcal{Y}^{in}) - P(y = \arg\max_k P^n(y = k|\mathbf{x} = \boldsymbol{x}; y \in \mathcal{Y}^{in})|\mathbf{x} = \boldsymbol{x}; y \in \mathcal{Y}^{in})] \\
&\quad \cdot P(\mathbf{x} = \boldsymbol{x}; y \in \mathcal{Y}^{in}).
\end{aligned} \tag{18}$$

- **Memorized case**:

$$\begin{aligned}
\Delta E_{\boldsymbol{x}} &= [\max_k P(y = k|\mathbf{x} = \boldsymbol{x}; y \in \mathcal{Y}^{in}) - \sum_{y^n \in \mathcal{Y}^{in}} P(y = y^n|\mathbf{x} = \boldsymbol{x}; y \in \mathcal{Y}^{in}) P^n(y = y^n|\mathbf{x} = \boldsymbol{x}; y \in \mathcal{Y}^{in})] \\
&\quad \cdot P(\mathbf{x} = \boldsymbol{x}; y \in \mathcal{Y}^{in}).
\end{aligned} \tag{19}$$

**Details on the derivation of error rate inflation (fig. 2)** Then, we describe the essential concepts depicted in fig. 2 in detail. For better clarity, we here restate the notations in section 3.4. We explicitly consider two specific sample points $\boldsymbol{x}_1$ and $\boldsymbol{x}_2$ being perturbed independently, corresponding to two different label noise modes. Let us assume its clean conditional probability as:

$$\begin{aligned}
P(y|\mathbf{x} = \boldsymbol{x}_1; y \in \mathcal{Y}^{all}) &= [p_1^1, ..., p_A^1, ..., p_{A+B}^1], \\
P(y|\mathbf{x} = \boldsymbol{x}_2; y \in \mathcal{Y}^{all}) &= [p_1^2, ..., p_A^2, ..., p_{A+B}^2],
\end{aligned} \tag{20}$$

and denote its noise transition matrix as $T^1 = \{T_{ij}^1\}_{i,j=1}^{A+B}$ and $T^2 = \{T_{ij}^2\}_{i,j=1}^{A+B}$, respectively. Here, $\{T_{ij}^1 = 0\}, \{T_{ij}^2 = 0\}$ for all $j > A$.

With eq. (1), we compute the corresponding noisy conditional probability for both samples as:

$$\begin{aligned}
P^n(y|\mathbf{x} = \boldsymbol{x}_1; y \in \mathcal{Y}^{all}) &= [\sum_{i=1}^{A+B} p_i^1 T_{i1}^1, ..., \sum_{i=1}^{A+B} p_i^1 T_{iA}^1, 0, ..., 0], \\
P^n(y|\mathbf{x} = \boldsymbol{x}_2; y \in \mathcal{Y}^{all}) &= [\sum_{i=1}^{A+B} p_i^2 T_{i1}^2, ..., \sum_{i=1}^{A+B} p_i^2 T_{iA}^2, 0, ..., 0].
\end{aligned} \tag{21}$$

Note that the *error rate inflation* is dependent on the *clean conditional probability over inlier classes*, *noisy conditional probability over inlier classes* and *sampling prior over inlier classes* as shown in eq. (18) and eq. (19).

Specifically, for sample $\boldsymbol{x}_1$, we have:

$$P(\mathrm{y} = k | \mathbf{x} = \boldsymbol{x}_1; \mathrm{y} \in \mathcal{Y}^{in}) = \frac{P(\mathrm{y} = k | \mathbf{x} = \boldsymbol{x}_1; \boldsymbol{y} \in \mathcal{Y}^{all})}{\sum_{i \in \mathcal{Y}^{in}} P(\mathrm{y} = i | \mathbf{x} = \boldsymbol{x}_1; \mathrm{y} \in \mathcal{Y}^{all})} = \frac{p_k^1}{\sum_{i=1}^{A} p_i^1},$$

$$P^n(\mathrm{y} = k | \mathbf{x} = \boldsymbol{x}_1; \mathrm{y} \in \mathcal{Y}^{in}) = \frac{P^n(\mathrm{y} = k | \mathbf{x} = \boldsymbol{x}_1; \boldsymbol{y} \in \mathcal{Y}^{all})}{\sum_{i \in \mathcal{Y}^{in}} P^n(\mathrm{y} = i | \mathbf{x} = \boldsymbol{x}_1; \mathrm{y} \in \mathcal{Y}^{all})} = \sum_{i=1}^{A+B} p_i^1 T_{ik}^1,$$

$$P(\mathbf{x} = \boldsymbol{x}_1; \mathrm{y} \in \mathcal{Y}^{in}) = \frac{\sum_{y \in \mathcal{Y}^{in}} P(\mathbf{x} = \boldsymbol{x}_1, \mathrm{y} = y; \mathrm{y} \in \mathcal{Y}^{all})}{\int \sum_{y \in \mathcal{Y}^{in}} P(\mathbf{x} = \boldsymbol{x}, \mathrm{y} = y; \mathrm{y} \in \mathcal{Y}^{all}) d\boldsymbol{x}}$$

$$\propto \sum_{y \in \mathcal{Y}^{in}} P(\mathbf{x} = \boldsymbol{x}_1, \mathrm{y} = y; \mathrm{y} \in \mathcal{Y}^{all}) \tag{22}$$

$$\propto \sum_{y \in \mathcal{Y}^{in}} P(\mathrm{y} = y | \mathbf{x} = \boldsymbol{x}_1; \mathrm{y} \in \mathcal{Y}^{all}) P(\mathbf{x} = \boldsymbol{x}_1; \mathrm{y} \in \mathcal{Y}^{all})$$

$$\xrightarrow{P(\mathbf{x}=\boldsymbol{x}_1; \mathrm{y}\in\mathcal{Y}^{all})=P(\mathbf{x}=\boldsymbol{x}_2; \mathrm{y}\in\mathcal{Y}^{all})=\delta}$$

$$\propto \sum_{y \in \mathcal{Y}^{in}} P(\mathrm{y} = y | \mathbf{x} = \boldsymbol{x}_1; \mathrm{y} \in \mathcal{Y}^{all}) = \sum_{i=1}^{A} p_i^1.$$

Simply changing the subscript leads us to the formulations for sample $\boldsymbol{x}_2$. To summarize, wrapping the above together, we have:

$$P(\mathrm{y} | \mathbf{x} = \boldsymbol{x}; \mathrm{y} \in \mathcal{Y}^{in}) = [\frac{p_1}{\sum_{i=1}^{A} p_i}, ..., \frac{p_A}{\sum_{i=1}^{A} p_i}],$$

$$P^n(\mathrm{y} | \mathbf{x} = \boldsymbol{x}; \mathrm{y} \in \mathcal{Y}^{in}) = [\sum_{i=1}^{A+B} p_i T_{i1}, ..., \sum_{i=1}^{A+B} p_i T_{iA}], \tag{23}$$

$$P(\mathbf{x} = \boldsymbol{x}_1; \mathrm{y} \in \mathcal{Y}^{in}) = \sum_{i=1}^{A} p_i.$$

We here omit the sample subscript and abbreviate the proportional symbol for simplicity. With eq. (18), eq. (19) and eq. (23), we can then compute and compare $\Delta E_{\boldsymbol{x}}$ in both ***fitted case*** and ***memorized case***:

$$\boxed{\Delta E_{\boldsymbol{x}} = \max[p_1, ..., p_A] - p_{\arg\max[\sum_{i=1}^{A+B} p_i T_{i1}, ..., \sum_{i=1}^{A+B} p_i T_{iA}]} \quad (\textbf{\textit{Fitted case}})} \tag{24}$$

$$\boxed{\Delta E_{\boldsymbol{x}} = \max[p_1, ..., p_A] - \sum_{i=1}^{A}(p_i \cdot \sum_{j=1}^{A+B} p_j T_{ji}) \quad (\textbf{\textit{Memorized case}})} \tag{25}$$

# D   Full proof of theorem 3.5 and theorem 3.6

**Error rate inflation comparison *s.t.* same noise ratio** To ensure a fair comparison, in this work, we focus on the impact of different label noise given the same noise ratio - modifying $O_{\boldsymbol{x}}$ and $C_{\boldsymbol{x}}$ while analyzing the trend of $\Delta E_{\boldsymbol{x}}$. Specifically, for above mentioned $\boldsymbol{x}_1$ and $\boldsymbol{x}_2$, we further assume:

$$O_{\boldsymbol{x}_1} + C_{\boldsymbol{x}_1} = O_{\boldsymbol{x}_2} + C_{\boldsymbol{x}_2} = \delta. \tag{26}$$

which leads us to:

$$\sum_{i=A+1}^{A+B} p_i^1 + \sum_{i=1}^{A} \sum_{j=1, j\neq i}^{A} T_{ij}^1 p_i^1 = \sum_{i=A+1}^{A+B} p_i^2 + \sum_{i=1}^{A} \sum_{j=1, j\neq i}^{A} T_{ij}^2 p_i^2 \longrightarrow \sum_{i=1}^{A} T_{ii}^1 p_i^1 = \sum_{i=1}^{A} T_{ii}^2 p_i^2 \tag{27}$$

*Please note, here the clean conditional probability is considered as known and fixed, while eq. (27) restricts the values of the noise transition matrix $T^1$ and $T^2$, given specific clean conditional probability. We then analyze and compare the error rate inflation in both conditions.*

## D.1 Proof of theorem 3.5 — Open-set noise vs Closed-set noise

In this section, we try to compare open-set noise and closed-set noise. Without loss of generality, we consider:

$$O_{\boldsymbol{x}_1} > O_{\boldsymbol{x}_2}. \tag{28}$$

Intuitively speaking, sample $\boldsymbol{x}_1$ is more affected by open-set noise compared to sample $\boldsymbol{x}_2$, thus corresponding to the interested 'open-set noise'.

As clarified by the toy example in section 3.4.1, without extra regularizations, the noise transition matrix is not identifiable. *We thus consider a simple compromise situation - in most classification problems, the majority of samples (with a high probability) belong to a specific class exclusively with high probability.*

Let us denote:

$$a = \arg\max_i P(\mathrm{y} = i | \mathbf{x} = \boldsymbol{x}_1; \mathrm{y} \in \mathcal{Y}^{all})$$

and

$$b = \arg\max_i P(\mathrm{y} = i | \mathbf{x} = \boldsymbol{x}_2; \mathrm{y} \in \mathcal{Y}^{all}).$$

We assume :

$$p_a^1 \to 1, \{p_i^1 \to 0\}_{i \neq a}, p_b^2 \to 1, \{p_i^2 \to 0\}_{i \neq b},$$

and we have:

$$O_{\boldsymbol{x}_1} = \sum_{i=A+1}^{A+B} p_i^1, \; O_{\boldsymbol{x}_2} = \sum_{i=A+1}^{A+B} p_i^2.$$

With eq. (28), we easily infer that: $a \in \mathcal{Y}^{out}$ while $b \in \mathcal{Y}^{in}$. Intuitively speaking, $\boldsymbol{x}_1$ is an open-set noise, with its clean conditional probability concentrated on one of the outlier classes, and vice versa for $\boldsymbol{x}_2$.

With eq. (27), we further have:

$$\sum_{i=1}^{A} T_{ii}^1 p_i^1 \approx \sum_{i=1}^{A} T_{ii}^1 \times 0 \approx 0,$$

$$\sum_{i=1}^{A} T_{ii}^2 p_i^2 \approx \sum_{i=1, i \neq b}^{A} T_{ii}^2 \times 0 + T_{bb}^2 \times 1 \approx T_{bb}^2.$$

Thus we have: $T_{bb}^2 \approx 0$, which enables us to analyze and compare $\Delta E_{\boldsymbol{x}_1}$ and $\Delta E_{\boldsymbol{x}_2}$:

**Fitted case** In this case, according to eq. (24), we have:

$$
\begin{aligned}
\Delta E_{\boldsymbol{x}_1} &= \max[p_1^1, ..., p_A^1] - p_{\arg\max[\sum_{i=1}^{A+B} p_i^1 T_{i1}^1, ..., \sum_{i=1}^{A+B} p_i^1 T_{iA}^1]} \\
&< \max[p_1^1, ..., p_A^1] - \min[p_1^1, ..., p_A^1] \\
&\xrightarrow{p_a^1 \to 1, \{p_i^1 \to 0\}_{i \neq a}, a \in \mathcal{Y}^{out}} \\
&\approx 0,
\end{aligned}
\tag{29}
$$

$$
\begin{aligned}
\Delta E_{\boldsymbol{x}_2} &= \max[p_1^2, ..., p_A^2] - p_{\arg\max[\sum_{i=1}^{A+B} p_i^2 T_{i1}^2, ..., \sum_{i=1}^{A+B} p_i^2 T_{iA}^2]} \\
&\xrightarrow{[\sum_{i=1}^{A+B} p_i^2 T_{i1}^2, ..., \sum_{i=1}^{A+B} p_i^2 T_{iA}^2] \approx [T_{a1}^2, T_{a2}^2, ..., \overbrace{0}^{b}, ..., T_{aA}^2]} \\
&= p_b^2 - p_n^2 \\
&\xrightarrow{p_b^2 \to 1, \{p_i^2 \to 0\}_{i \neq b}, b \in \mathcal{Y}^{in}, n \neq b} \\
&\approx 1.
\end{aligned}
\tag{30}
$$

**Memorized case** In this case, according to eq. (25), we similarly have:

$$\Delta E_{\boldsymbol{x}_1} = \max[p_1^1, ..., p_A^1] - \sum_{i=1}^{A}(p_i^1 \cdot \sum_{j=1}^{A+B} p_j^1 T_{ji}^1) \approx 0, \tag{31}$$

$$\Delta E_{\boldsymbol{x}_2} = \max[p_1^2, ..., p_A^2] - \sum_{i=1}^{A}(p_i^2 \cdot \sum_{j=1}^{A+B} p_j^2 T_{ji}^2) \approx 1. \tag{32}$$

We wrap up above for theorem D.2:

**Theorem D.1** (Open-set noise *vs* Closed-set noise). *Let us consider sample $\boldsymbol{x}_1$, $\boldsymbol{x}_2$ fulfilling eq. (26) and eq. (28) - compared to $\boldsymbol{x}_2$, $\boldsymbol{x}_1$ is considered as more prone to open-set noise. Let us denote $a = \arg\max_i P(\mathbf{y} = i | \mathbf{x} = \boldsymbol{x}_1; \mathbf{y} \in \mathcal{Y}^{all})$ and $b = \arg\max_i P(\mathbf{y} = i | \mathbf{x} = \boldsymbol{x}_2; \mathbf{y} \in \mathcal{Y}^{all})$, we assume (with a high probability): $p_a^1 \to 1, \{p_i^1 \to 0\}_{i \neq a}$ and $p_b^2 \to 1, \{p_i^2 \to 0\}_{i \neq b}$. Then, we have:*

$$\Delta E_{\boldsymbol{x}_1} < \Delta E_{\boldsymbol{x}_2}$$

*in both **fitted case** and **memorized case**.*

## D.2 Derivation of theorem 3.5 — 'hard' open-set noise vs 'easy' open-set noise

In this part, we try to analyze and compare 'hard' open-set noise with 'easy' open-set noise. For better clarification, we repeat here the essential statements:

$$T_{out}^1 = T^{easy} = \begin{bmatrix} \frac{1}{A} & \cdots & \frac{1}{A} \\ \cdots & \cdots & \cdots \\ \frac{1}{A} & \cdots & \frac{1}{A} \end{bmatrix}_{B \times A} \tag{33}$$

and

$$T_{out}^2 = T^{hard} = \begin{bmatrix} 0 & \cdots & 1 \\ \cdots & \cdots & \cdots \\ 1 & \cdots & 0 \end{bmatrix}_{B \times A} \tag{34}$$

and

$$T_{in}^1 = T_{in}^2 = \mathbf{I}. \tag{35}$$

Especially, for $T^{easy}$, we have $T_{ij} = \frac{1}{A}$ everywhere; for $T^{hard}$, we denote as $H_i : \{\arg_j(T_{ji}^{hard} = 1)\}_{i=1}^{A}$ the set of corresponding outlier classes $j \in \mathcal{Y}^{out}$ confused to inlier class $i \in \mathcal{Y}^{in}$. We also have:

$$[p_1^1, ..., p_A^1, ..., p_{A+B}^1] = [p_1^2, ..., p_A^2, ..., p_{A+B}^2]$$

.

**Fitted case** In this case, according to eq. (24), for sample $\boldsymbol{x}_1$ with 'easy' open-set noise, we have:

$$\begin{aligned}
\Delta E_{\boldsymbol{x}_1} &= \max[p_1^1, ..., p_A^1] - p_{\arg\max[\sum_{i=1}^{A+B} p_i^1 T_{i1}^1, ..., \sum_{i=1}^{A+B} p_i^1 T_{iA}^1]} \\
&= \max[p_1^1, ..., p_A^1] - p_{\arg\max[p_1^1 + \frac{1}{A}\sum_{i=A+1}^{A+B} p_i^1, ..., p_A^1 + \frac{1}{A}\sum_{i=A+1}^{A+B} p_i^1]} \\
&= 0,
\end{aligned} \tag{36}$$

and, for sample $\boldsymbol{x}_2$ with 'hard' open-set noise, we have:

$$\begin{aligned}
\Delta E_{\boldsymbol{x}_2} &= \max[p_1^2, ..., p_A^2] - p_{\arg\max[\sum_{i=1}^{A+B} p_i^2 T_{i1}^2, ..., \sum_{i=1}^{A+B} p_i^2 T_{iA}^2]} \\
&= \max[p_1^2, ..., p_A^2] - p_{\arg\max[p_1^2 + \sum_{b \in H_1} p_b^2, ..., p_A^2 + \sum_{b \in H_A} p_b^2]} \\
&\in [0, \ \max[p_1^2, ..., p_A^2] - \min[p_1^2, ..., p_A^2]].
\end{aligned} \tag{37}$$

**Memorized case** In this case, according to eq. (25), for sample $\boldsymbol{x}_1$ with 'easy' open-set noise, we have:

$$
\begin{aligned}
\Delta E_{\boldsymbol{x}_1} &= \max[p_1^1, ..., p_A^1] - \sum_{i=1}^{A}(p_i^1 \cdot \sum_{j=1}^{A+B} p_j^1 T_{ji}^1) \\
&= \max[p_1^1, ..., p_A^1] - \sum_{i=1}^{A} p_i^1(p_i^1 + \frac{1}{A}\sum_{i=A+1}^{A+B} p_i^1).
\end{aligned}
\tag{38}
$$

and, for sample $\boldsymbol{x}_2$ with 'hard' open-set noise, we have:

$$
\begin{aligned}
\Delta E_{\boldsymbol{x}_2} &= \max[p_1^2, ..., p_A^2] - \sum_{i=1}^{A}(p_i^2 \cdot \sum_{j=1}^{A+B} p_j^2 T_{ji}^2) \\
&= \max[p_1^2, ..., p_A^2] - \sum_{i=1}^{A} p_i^2(p_i^2 + \sum_{j \in H_i} p_j^2)
\end{aligned}
\tag{39}
$$

We further have:

$$
\Delta E_{\boldsymbol{x}_1} - \Delta E_{\boldsymbol{x}_2} = \sum_{i=1}^{A} p_i^1(\sum_{j \in H_i} p_j^1 - \frac{1}{A}\sum_{i=A+1}^{A+B} p_i^1).
$$

Let $a_i = p_i^1, b_i = \sum_{j \in H_i} p_j^1 - \frac{1}{A}\sum_{i=A+1}^{A+B} p_i^1$, we have:

$$
\Delta E_{\boldsymbol{x}_1} - \Delta E_{\boldsymbol{x}_2} = \sum_{i=1}^{A} a_i b_i.
$$

To summarize, we wrap up the above together:

**Theorem D.2** ('Hard' open-set noise *vs* 'easy' open-set noise)**.** *Let us consider sample $\boldsymbol{x}_1$, $\boldsymbol{x}_2$ fulfilling eq. (26) and eq. (11). We set the corresponding noise transition matrix as in eq. (33), eq. (34) and eq. (35). We further assume $[p_1^1, ..., p_A^1, ..., p_{A+B}^1] = [p_1^2, ..., p_A^2, ..., p_{A+B}^2]$. Then, we have:*

$$
\Delta E_{\boldsymbol{x}_1} \leq \Delta E_{\boldsymbol{x}_2}
$$

*in **fitted case**,*

$$
\Delta E_{\boldsymbol{x}_1} - \Delta E_{\boldsymbol{x}_2} = \sum_{i=1}^{A} a_i b_i
$$

*in **memorized case**. Here, $a_i = p_i^1, b_i = \sum_{j \in H_i} p_j^1 - \frac{1}{A}\sum_{i=A+1}^{A+B} p_i^1$.*

**Theorem D.3** (Rearrangement Inequality)**.** *For the sequences $a_1, a_2, \ldots, a_n$ and $b_1, b_2, \ldots, b_n$, where $a_1 \leq a_2 \leq \ldots \leq a_n$ and $b_1 \leq b_2 \leq \ldots \leq b_n$, the rearrangement inequality is given by:*

$$
a_1 \cdot b_1 + a_2 \cdot b_2 + \ldots + a_n \cdot b_n \geq a_1 \cdot b_{\sigma(1)} + a_2 \cdot b_{\sigma(2)} + \ldots + a_n \cdot b_{\sigma(n)} \geq a_1 \cdot b_n + a_2 \cdot b_{n-1} + \ldots + a_n \cdot b_1
$$

*Here, $\sigma$ denotes a permutation of the indices $1, 2, \ldots, n$. The leftmost expression corresponds to the case where $\sigma(i) = i$ (identity permutation), and the rightmost expression corresponds to the case where $\sigma(i) = n + 1 - i$ (reverse permutation).*

# E  Revisiting LNL methods

## E.1  Revisiting existing LNL methods with open-set noise

In this section, we further investigate the learning effectiveness of existing LNL methods on previously discussed open-set label noise, especially the dominant ones based on sample selection - these methods often integrate different regularization terms and off-the-shelf techniques, resulting in state-of-the-art performance. In essence, such methods typically include a sample selection module along with a

robust training module. Here, we briefly denote the clean subset selected by the original method as $X_{clean}$ and denote the entire dataset as $X_{all}$. Moreover, we consider integrating the previously mentioned open-set detection mechanism into current LNL methods - we denote as $X_{in}$ an inlier subset based on entropy dynamics. Then, maintaining the robust training module unchanged, we consider below three different variants (the involved LNL method abbreviated as **X**, the inlier subset detection method abbreviated as **EntSel**):

- X: Robust training using $X_{clean}$, i.e., the original method;

- EntSel: Robust training using $X_{in}$;

- X + EntSel: Robust training using $X_{in} \cap X_{clean}$.

Specifically, we test with two representative LNL methods with well-maintained open-source implementations: SSR [5] and DivideMix [11]. Please refer to appendix E.2 for more details. In fig. 6, we show results on CIFAR100-O and ImageNet-O.

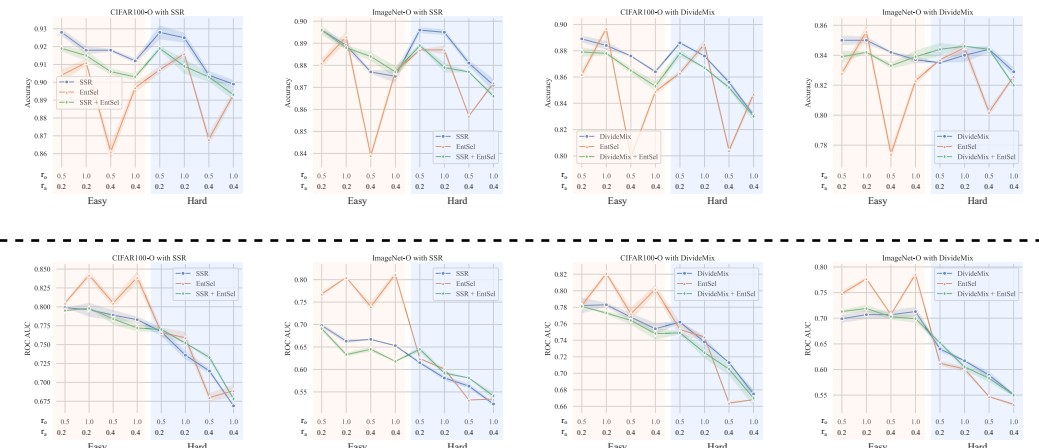

Figure 6: Evaluation of directly supervised training with different noise modes/ratios. First row: Closed-set classification accuracy; Second row: Open-set detection ROC AUC.

First, focusing on the classification accuracy of the model, we observe that 1) using EntSel instead of the original method leads to a reduction in classification accuracy in the mixed noise scenario (SSR vs EntSel and DivideMix vs EntSel); in pure open-set noise only scenarios, there are no obvious trends showing differences in different variant models. 2) the classification accuracy for mixed noise is significantly lower than that of only open-set noise at the same noise ratio, which further confirms that closed-set noise is more harmful than open-set noise.

Furthermore, we demonstrate the performance of this model in detecting open-set samples - the introduction of EntSel significantly enhances the effectiveness of open-set detection, especially when the open-set noise is set to 'easy' mode. This also further confirms our theoretical analysis in section 3.5 and experimental results in section 4.2.

Table 1: Results on WebVision dataset.

| Method | Accuracy (%) | ROC AUC (%) |
|---|---|---|
| SSR | **77.48** | 80.84 |
| EntSel | 77.08 | **85.43** |
| SSR + EntSel | 76.04 | 79.90 |
| DivideMix | **74.08** | **86.39** |
| EntSel | 62.96 | 81.66 |
| DivideMix + EntSel | 58.94 | 83.85 |

We report results for the WebVision dataset in table 1, reaffirming that combining 'EntSel' with 'SSR' significantly enhances open-set detection performance. Notably, most open-set noise in WebVision seems to arise from factors like text co-occurrence rather than semantic similarity, categorizing it more as 'easy' open-set noise. This may explain why EntSel effectively improves open-set detection in this context. However, when combining EntSel with DivideMix, both classification accuracy and open-set detection decrease, indicating that the robustness of the EntSel method itself is questionable. Additionally, simply merging SSR/DivideMix with EntSel using subset intersection (X + EntSel) also leads to a decrease in both classification accuracy and open-set detection performance. Finally, it's worth mentioning that, despite having lower classification accuracy than SSR, DivideMix outperforms SSR in open-set detection ROC AUC scores. All above illustrates that simply evaluating the classification accuracy may be one-sided.

## E.2 Details of involved methods

**DivideMix** [11] Denoting as $\mathcal{L} = \{l_i\}_{i=1}^N$ the losses of all samples, DivideMix proposes to model it (after min-max normalization) with a Gaussian Mixture Model. The probabilities $\{p_i\}_{i=1}^N$ of each sample belonging to the component with a smaller mean value are then extracted. Samples with probability $p_i$ greater than the threshold $\theta$ are then identified as a "clean" subset. Link to code: https://github.com/LiJunnan1992/DivideMix.

**SSR** [5] In contrast to DivideMix, SSR extracts features for each sample and constructs a neighbourhood graph. By computing the nearest neighbour labels for each sample, a pseudo-label distribution $\boldsymbol{p}$ is obtained through a KNN voting process. The consistency $c = \boldsymbol{p}_y/\boldsymbol{p}_{max}$ between this voted distribution and the given noisy label $y$ (logit label) is then calculated. Samples with consistency $c$ greater than the threshold $\theta$ are identified as part of the "clean" subset. Link to code: https://github.com/MrChenFeng/SSR_BMVC2022.

**EntSel** We also provide a concise overview of the steps involved in EntSel, following a methodology similar to DivideMix. Denoting as $\mathcal{E} = \{e_i\}_{i=1}^N$ the entropy of all samples' predictions, we similarly model it (after min-max normalization) with a Gaussian Mixture Model. The probabilities $\{p_i\}_{i=1}^N$ of each sample belonging to the component with a smaller mean value are then extracted. Samples with probability $p_i$ greater than the threshold $\theta'$ are then identified as "inlier" subset.

Generally, we have a closed-set classifier g and an encoder f, and we use it for training based on the selected subset. Existing sample selection methods usually rely on an estimated prediction and a threshold to help filter clean samples. Our proposed OpenAdaptor focuses on the difference between open-set and closed-set samples. When integrating them, we propose two different strategies: absorption and exclusion.

## E.3 Implementation details

**Experiment details** For both SSR and DivideMix, we employ model and optimization configurations on the same dataset. Specifically, for CIFAR100-O and ImageNet-O, we utilize the Pres-ResNet18 model, trained for 300 epochs with a batch size of 128 and a learning rate of 0.02, and a cosine annealing schedule was implemented. For the WebVision dataset, we utilize the ResNet18 model, training for 120 epochs with a reduced batch size of 32. The learning rate is set to 0.01 and controlled by a cosine annealing scheduler too. Additionally, a warm-up training phase of 10 epochs is implemented in the CIFAR100-O and ImageNet-O experiments, while a 5-epoch warm-up training phase is utilized in the WebVision experiment.

**Hyperparameters** In all experiments, we set the sample selection threshold $\theta' = 0.5$ for EntSel. For SSR, we employ a sample selection threshold $\theta = 1.0$ in all experiments. For DivideMix, the sample selection threshold remains constant at $\theta = 0.5$ across all experiments. Both SSR and DivideMix incorporate MixUp, and we adhere to the original paper's choices by setting the MixUp coefficient to 4 for experiments on CIFAR100-O and ImageNet-O and to 0.5 for experiments on WebVision. Please note, as exploring and comparing these methods are not our focus, we believe there exist better hyperparameter settings.

**Robustness of EntSel**     A smaller $\theta'$ for EntSel leads to better performance on WebVision - especially when EntSel is used with DivideMix. If we set $\theta' = 0.2$, our classification accuracy increases from 62.96% to 67.2%, ROC AUC increases from 0.8166 to 0.8599 (table 1). However, we are keen to use fixed hyperparameters in all experiments as we emphasize that the hyperparameter robustness is also critical for LNL methods.

## F    More examples of open-set noise in WebVision dataset

In this section, we present additional examples of open-set noise within the 'Tench' class of the WebVision dataset. We trace the origin of web pages containing some open-set noise images. Remarkably, we identify that the appearance of the term 'Tench' or related keywords is prevalent on the web pages hosting these open-set noise images. We posit that this occurrence is attributed to the data collection process on the web. Specifically, in the course of keyword searches and crawling for images, instances were inadvertently included due to the presence of keywords in image descriptions or accompanying text, such as people with 'tench' in the name, or related fishing tools. As highlighted earlier, the prevalent belief in the current LNL community is that real-world noise primarily arises from confusion induced by semantic similarity. Consequently, numerous recent studies have concentrated on instance-dependent noise and related theoretical analysis. ***However, our findings here indicate that in real-world scenarios, particularly in web-crawled datasets, noise may be unrelated to semantics but instead caused by other latent high-dimensional information, such as accompanying text here. Addressing such real-world noise requires increased attention and further exploration.***

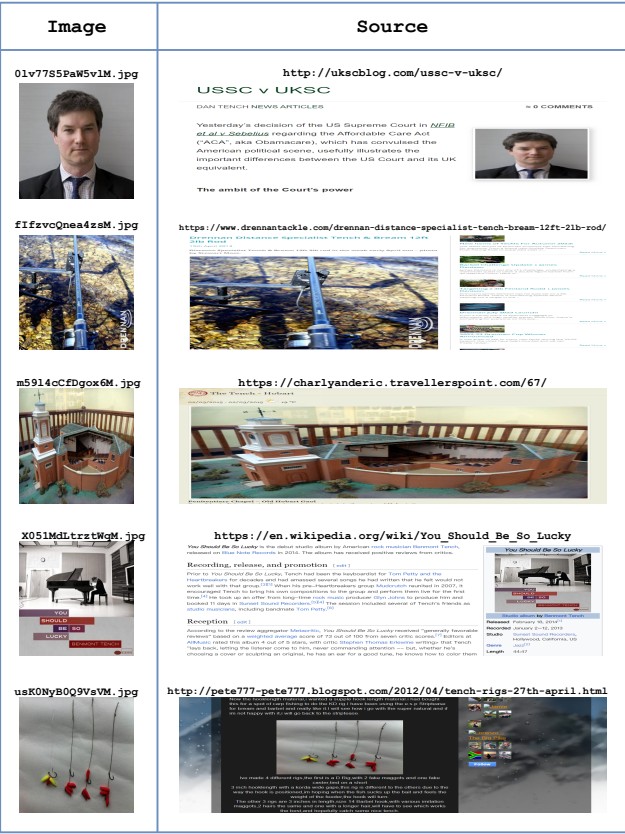

Figure 7:   Open-set noise examples in class 'Tench' of WebVision dataset with path: `/google/q0001/`. The source images are resized to fit the layout. Please note that the web links here are obtained in May 2024, and there is no guarantee that they will always be valid in the future.

