# OpenReview forum: "Rethinking Open-set Noise in Learning with Noisy Labels"
_NeurIPS.cc/2024/Conference — Submitted to NeurIPS 2024_

### Official Review · Reviewer_AwyQ · 2024-06-12

**Soundness:** 3
**Presentation:** 2
**Contribution:** 2
**Rating:** 4
**Confidence:** 4

**Summary:**

The paper extends the problem setting of learning with noisy labels (LNL) to include open-set noise, where noisy labels may come from unknown categories, in contrast to the traditional focus on closed-set noise. The authors theoretically compare the impacts of open-set and closed-set noise and analyze detection mechanisms based on prediction entropy. They construct two open-set noisy datasets, CIFAR100-O and ImageNet-O, and introduce an open-set test set for the WebVision benchmark to validate their findings. Their results show that open-set noise exhibits distinct characteristics from closed-set noise. The paper emphasizes the need for comprehensive evaluation methods for models in the presence of open-set noise, calling for further research in this area.

**Strengths:**

- The research problem is interesting. Compared with learning with closed-set noise, learning with open-set noise is under-explored.
- The theoretical analysis seems to be solid.

**Weaknesses:**

- Some technical details are hard to follow. Writing needs to be polished.
- The contribution from the algorithm perspective is not enough.

**Questions:**

- I am a bit confused about the title of this work. This paper provides some insights into the research community. However, it does not provide some essentially different and surprising conclusions. Therefore, perhaps it is not suitable to use "rethinking".
- The review part is not comprehensive. The methods combining unsupervised and semi-supervised learning are not reviewed and discussed.
- The assumption that the noisy labeling will not affect the sampling prior is a bit strong. Could the paper supplement more intuitions about this assumption?
- Will the size of label space for the open-set label, i.e., $B$, affect the theoretical analysis?
- There is a gap between Line 165 and the following explanations, which makes it hard to understand.
- Does Eq. (7) mean the bounded noise?
- The method of predicting entropy values is similar to the methods in OOD detection. Could the method supplement more details?
- The paper does not provide enough contribution from the algorithm perspective, such as how to estimate the transition matrix and how to better use/remove the open-set data for better model robustness. The harm of open-set label noise is less than closed-set one is known, which has been reflected in the previous works (results or related discussions).

**Limitations:**

N/A.

---

> ### Author Rebuttal · Authors · 2024-08-06
>
> Thank you very much for your careful review. We sincerely appreciate your time and effort in reading our paper, as well as the insightful and constructive feedback.
> >*Q1: Some technical details …needs to be polished.*
>
> A1: We will make further revisions to the manuscript to improve clarity. We would like to kindly note that the other two reviewers, **gSVD** and **6bHU**, agreed that the paper was generally clear and well-written. We would be very happy to clarify any specific parts the reviewer points out.
>
> >*Q2: The contribution … is not enough.*
>
> A2: Please refer to A10 for *Q10*.
>
> >*Q3: I am a bit confused about the title of this work. ... perhaps it is not suitable to use "rethinking".*
>
> A3: Thanks for the suggestion. We would be happy to drop the term ‘rethinking’.
>
> >*Q4: The review part is not comprehensive...*
>
> A4: We would kindly bring attention to the fact that we have cited many papers based on semi-supervised and unsupervised techniques in the current version of the related works section (L67-L69 in paper). Since most of these methods aim to achieve better baseline results by introducing more techniques, and our paper focuses on the theoretical analysis of open-set noise, we thus did not provide detailed descriptions of the methods before. Following the reviewer's suggestion, we will divide the related work section into three parts in the updated manuscript: *Statistical-Consistent Methods*, *Statistical-Approximate Methods*, and *Exploration of Open-Set Noise*. The second part will include introduction to the works referred to by the reviewer. If the reviewer has more references to suggest, we would be very happy to include them.
>
> >*Q5: The assumption that the noisy labeling ... more intuitions about this assumption?*
>
> A5: We would like to kindly note that most works in the LNL (Learning with Noisy Labels) area adhere to this assumption. The change in sampling prior ($P(x)$) is often known as the covariate shift problem—this falls beyond the scope of this work and LNL. Furthermore, if the assumption is that both labelling and sampling prior change, the problem becomes nearly intractable.
>
> >*Q6: Will the size of label space for the open-set label, i.e., B, affect the theoretical analysis?*
>
> A6: Thanks for the question. We are happy to clarify further that value of $B$ will not affect the theoretical analysis.
>
> >*Q7: There is a gap between Line 165 and the following explanations...*
>
> A7: Thank you for the comment. In the updated manuscript, following the suggestion we will include the explanation of each case, at the bullet point that it is defined. For example,
>
> - Memorized case:  the model completely memorises the noisy labels: $P^f(y|x=x; y\in \mathcal{Y}^{in}) = P^{y^n}(y|x=x; y\in \mathcal{Y}^{in})$; This, could arise in the scenarios that a high capacity model is overfitted on a small, noisy dataset.
>
> >*Q8: Does Eq. (7) mean the bounded noise?*
>
> A8: Here $\delta$ is a notation to denote the noise ratio without any further assumptions on it. Clearly 0$\leq$delta$\leq$1, that is the noise ratio is bounded between 0 and 1, however, the main purpose of Eq.(7) is to convey that $x_1$ and $x_2$ have overall the same noise levels. This is needed for a fair comparison of different noise given the same noise ratio.
>
> >*Q9: The method of predicting entropy ... Could the method supplement more details?*
>
> A9: Indeed, as the reviewer mentioned, entropy-based detection techniques have been widely used in out-of-distribution (OOD) detection (e.g., [1], [2]). Previous LNL methods have also adopted similar approaches for open-set detection (discussed in L250-252 of the paper). We would like to kindly note, that we do not claim entropy-based open-set detection as our contribution; rather, we analyze its applicability to the two different open-set noise we proposed. Following the reviewer's comment, we would be happy to revisit here the steps of the entropy-based sample selection we used [L603-L607 in the appendix]:
>
> *"Denoting as $e_i, i=1,...,N$ the entropy of all samples' predictions, we model it (after min-max normalization) with a GMM. The probability $p_i, 1,...,N$ of each sample belonging to the component with a smaller mean value are then extracted. Samples with probability $p_i$ greater than a threshold are then identified as ''inlier" subset."*
>
> We will update the manuscript to move above to the main content.
>
> [1] Chan et al. "Entropy maximization and meta classification for out-of-distribution detection in semantic segmentation." ICCV. 2021.
>
> [2] Xing et al. "Learning by Erasing: Conditional Entropy Based Transferable Out-of-Distribution Detection." AAAI. 2024.
>
>
> >*Q10: The paper does not provide ...  to estimate the transition matrix ... remove the open-set data... The harm of open-set label noise is less than closed-set one is known...*
>
> A10: We respectfully disagree with the reviewer's argument. We want to clarify that our focus is not to explore better ways to estimate the open-set noise transition matrix nor investigate how to detect open-set noise samples from the dataset more effectively. As the reviewer noted, "The harm of open-set label noise is less than closed-set one is known" - However, previous work mostly focused on ‘easy’ open-set noise in the memorized case. In addition to confirming this *theoretically*, we further demonstrate that the newly proposed ‘hard’ open-set noise also shows less harm than closed-set noise. More importantly, we compare 'hard' open-set noise versus 'easy' open-set noise, finding their opposite trends across the two cases. In the *fitted case*, 'easy' open-set noise is less harmful, whereas in the *memorized case*, the impact of 'hard' open-set noise is comparatively smaller. Due to space limitation, we kindly refer the reviewer to our paper and general response for more detailed clarification of contributions.
>
> Thanks once again to the reviewer. If there are any further questions, we would be happy to discuss them at the next stage.

---

> > ### Comment · Reviewer_AwyQ · 2024-08-13
> >
> > Thank you for the feedback. The response addresses some concerns. There are still some problems in the current form. First, for A5, please refer to [1] to check the discussion about covariate shifts in LNL. This does not fall beyond the scope of LNL. Second, for A9, the reviewer just needs some explanations about the relation between the work and OOD detection work to address/highlight the technical contributions of this work, which does involve or affect the contribution of this paper. Third, the paper does not provide enough contribution from the algorithm perspective, which is further confirmed. I understand the analysis of hard and easy open-set noise and do not deny them. This is just because even if there is such noise, how do we detect and handle/use them? It is not solved.
> >
> > I appreciate the author's response. Due to the above points, I may not be convinced now. When the reviewers and AC discuss, I am willing to listen to other views. Although I am slightly negative, I am not opposed to accepting this paper.
> >
> > [1] Confidence Scores Make Instance-dependent Label-noise Learning Possible. ICML 2021.

---

> ### Author Response · Authors · 2024-08-13
>
> Many thanks for the reviewer's detailed feedback. We are glad the reviewer is not opposed to accepting the paper. We understand and respect the reviewer’s comments. Here, we would like to further clarify remaining concerns, for your optional reference:
>
> - Regarding covariate shift, we appreciate the reviewers' recommended reference. We have carefully read [1] and would like to provide further clarification. The 'covariate shift' mentioned in [1] refers to distribution change between the selected subset and the original training set due to the sample selection (section 2.2), i.e., $P_{\text{selected}}(x) \neq P_{\text{train}}(x)$. For example, sample selection methods based on 'small-loss' mechanism tend to choose more samples with smaller losses (possibly images with more explicit class-related visual signals), leading to potential distributional changes. We would like to note, similar to most methods based on noise transition matrix, the method proposed in [1] also assumes $P_{\text{train}}(x) = P_{\text{test}}(x)$ implicitly.
>
> - Regarding the relationship between our work and out-of-distribution (OOD) detection, we would be pleased to include more discussions in the updated related work section. We want to clarify that we do not claim the entropy-based open-set detection method as our contribution; rather, we analyzed its applicability under different open-set noise conditions to show the possible limitations of existing LNL techniques towards different open-set noise. Moreover, our method can also provide interesting insights to current OOD methods. For example, the different tendency of ‘hard’ open-set noise and ‘easy’ open-set noise in different cases (*fitted* vs. *memorized*) may also help to guide current OOD methods in dealing with different OOD samples.
>
> - Regarding the contributions of this paper, we would like to reiterate that our focus is to investigate and compare different open-set noises and their impacts. We would also kindly refer to the general response for our detailed contributions. Considering how more advanced sample selection frameworks could be used to detect them or better estimate the noise transition matrix is an intriguing direction for future exploration, but it is beyond the scope of this work. For example, as reviewer *3Ttr* mentioned, our findings could be used to guide such approaches. We believe that we make a solid contribution to the LNL community.
>
>
> [1] Confidence Scores Make Instance-dependent Label-noise Learning Possible. ICML 2021.

---

### Official Review · Reviewer_gSVD · 2024-07-08

**Soundness:** 3
**Presentation:** 2
**Contribution:** 3
**Rating:** 6
**Confidence:** 4

**Summary:**

This paper introduces an approach to address the challenge of open-set noise in the context of learning from noisy labels. The authors propose a method that differentiates between 'easy' and 'hard' types of open-set noise, which is critical for improving the robustness and performance of learning models faced with noisy data. By integrating existing Learning with Noisy Labels (LNL) techniques with novel entropy-based noise detection mechanisms, the paper presents both theoretical insights and empirical validations of the proposed methods. The contributions are significant as they offer a refined perspective on handling different noise complexities, which can enhance the utility of machine learning models in real-world applications dealing with noisy labels.

**Strengths:**

Originality: The paper addresses the issue of open-set noise in learning from noisy labels with a novel approach, differentiating between 'easy' and 'hard' noise types. This nuanced consideration is original as it pushes the boundaries of how noise is typically treated in noisy label learning.

Quality: The theoretical explanations are thorough and complemented by robust empirical evidence that strengthens the methodological claims.

Clarity: The paper is well-structured, offering clear explanations of complex concepts, which aids in understanding the proposed methods and their implications.

Significance: The significance of this work is evident as it tackles a critical issue that can potentially enhance model robustness and performance in real-world scenarios where label noise is common.

**Weaknesses:**

Dependency on Specific Methods: The reliance on entropy-based techniques for noise distinction may not generalize across all scenarios or noise types.

Experimental Scope: The experiments primarily utilize synthetic datasets, which might not fully capture the complexity of real-world data applications.

**Questions:**

Regarding Open-set Noise Types: How do 'easy' and 'hard' open-set noises specifically impact the robustness of different deep learning models across various architectures?

**Limitations:**

Limited Experimental Scope: The experimental validation focuses predominantly on synthetic datasets like CIFAR100-O and ImageNet-O. While these are commonly used in the research community for benchmarking, the real-world implications of the findings might be limited without additional testing on more varied and real-world datasets.

---

> ### Author Rebuttal · Authors · 2024-08-06
>
> Thank you very much for your careful review. We sincerely appreciate your time and effort in reading our paper, as well as the insightful and positive feedback.
>
> > *Q1: Dependency on Specific Methods: The reliance on entropy-based techniques for noise distinction may not generalize across all scenarios or noise types.*
>
> A1: We would like to thank the reviewer for the comment. Indeed, entropy-based techniques may not be the ultimate answer to the problem of learning under open-set noise, however, we show its applicability and usefulness in the paper, including its limitations (see Section 3.5). We feel, that overall our work advances the state of the art by:
> - Extending the noise transition matrix concept to address open-set noise and reformulating the learning with noisy labels (LNL) problem.
> - Analyzing two cases for offline evaluation: the fitted case (model fits the noisy distribution) and the memorized case (model memorizes the noisy labels).
> - Confirming that open-set noise has less impact than closed-set noise and noting opposite trends between 'easy' and 'hard' open-set noise in these cases.
> - Proposing an additional open-set detection task and conducting preliminary experiments, given the small impact of open-set noise on classification performance.
> - Analyzing an entropy-based detection mechanism, which is effective mainly for 'easy' open-set noise.
> - Creating two synthetic datasets, CIFAR100-O and ImageNet-O, and introducing a new open-set test set for WebVision to facilitate controlled experiments.
>
> > *Q2: Experimental Scope: The experiments primarily utilize synthetic datasets, which might not fully capture the complexity of real-world data applications.*
>
> A2: Thank you for the feedback. To conduct comparative experiments with controllable noise ratios, we introduced two novel open-set noise datasets: ImageNet-O and CIFAR100-O. Additionally, we conducted experiments on the real-world dataset WebVision, as detailed in Appendix E. We will update the manuscript to clarify this clearer.
>
> > *Q3: Regarding Open-set Noise Types: How do 'easy' and 'hard' open-set noises specifically impact the robustness of different deep learning models across various architectures?*
>
> A3: In Sections 3.4.2 and 3.5 of the paper, we explore and compare the effects of simple and hard open-set noise in two different cases on model classification accuracy. We would be happy to further clarify: the impact of 'hard' open-set noise and 'easy' open-set noise shows an opposite trend in two different cases. In the *fitted case*, 'easy' open-set noise is less harmful, whereas in the *memorized case*, the impact of 'hard' open-set noise is comparatively smaller.
>
> > *Q4: Limited Experimental Scope: The experimental validation focuses predominantly on synthetic datasets like CIFAR100-O and ImageNet-O. While these are commonly used in the research community for benchmarking, the real-world implications of the findings might be limited without additional testing on more varied and real-world datasets.*
>
> A4: Please refer to A2 for *Q2*.
>
>
> Thanks once again to the reviewer. If there are any further questions, we would be happy to discuss them at the next stage.

---

### Official Review · Reviewer_3Ttr · 2024-07-12

**Soundness:** 3
**Presentation:** 2
**Contribution:** 2
**Rating:** 5
**Confidence:** 3

**Summary:**

This paper focuses on open-set label noise problem. Authors first formally extending closed-set transition matrix to open-set transition matrix and define two noise ratios for open-set and closed-set separately.
Then authors define error inflation rate as a measurement for noisy label impact and measure for two conditions, classifier fitted noisy distribution or memorized (overfit) noisy label. Later, authors propose a new type of open-set noise by exclusively transitioning outlier classes to a specific inlier class, and consider this as a "hard" open-set noise and traditional open-set noise as "easy" case. Authors further analysis two noise types on two classifier conditions and claim traditional entropy based open-set detection might only works on "easy" case. Experiments are performed on CIFAR-100, ImageNet and Webvision datasets.

**Strengths:**

Authors formally define open-set noise with a similar symmetric/asymmetric setup as closed-set noise, and find out that it shows opposite trend with different classifier cases.

**Weaknesses:**

- The experiment parts lack of baselines. With a new type of noise proposed, previous baselines on easy open-set noise should be run to assess the performance gap and set up the benchmark.
- Figure 4 (a) and (b) have similar distribution, it is hard to draw conclusions from entropy dynamics.
- Supp E.1 results are confusing. "X+EntSel" should be a better strategy since it selects inlier clean samples. However, why the closed-set classification accuracy is always the worst? Table 1 Webvision result is similar as well. Why is the claim "EntSel + SSR improves open-set detection performance" valid? The Acc and AUC are both dropping after adding EntSel. Why SSR/DivideMix + EntSel is always the worst performance? Considering it is a combination of inlier and clean, shouldn't it be the best performing one? I assume this is still the normal accuracy and AUC, which is the higher the better.

**Questions:**

- what is the pre-trained weight fitted case used?

**Limitations:**

Authors do not address any limitations in conclusion. A possible limitation might be related to approximation of fitted and memorized classifiers.

---

> ### Author Rebuttal · Authors · 2024-08-06
>
> Thanks very much for the careful review. We sincerely appreciate your time and effort in reading our paper, as well as the insightful and constructive feedback.
>
> >*Q1: The experiment parts lack of baselines. ..., previous baselines on easy open-set noise ... set up the benchmark.*
>
> A1: Thanks very much for the suggestion. We note that, in the Appendix E, we have included results for the DivideMix and SSR methods on the newly proposed dataset. Following the reviewer’s suggestion, we include results in ***Table. 3*** with two additional methods—EvidentialMix [1], DSOS [2] —on the CIFAR100-O dataset below.
>
> | Method / Noise Ratio | 0.2 Easy | 0.4 Easy | 0.2 Hard | 0.4 Hard |
> |---|---|---|---|---|
> | SSR | 0.889 | 0.875 | 0.895 | 0.871 |
> | DivideMix | 0.783 | 0.754 | 0.738 | 0.675 |
> | EvidentialMix[1] | 0.884 | 0.827 | 0.898 | 0.872 |
> | DSOS[2] | 0.846 | 0.765 | 0.854 | 0.832 |
> **Table. 3 Results on CIFAR100-O**
>
> We can generally agree that different methods exhibit varying sensitivities to open-set noise. Please note that analyzing the performance of these methods is not straightforward and requires further exploration, as they often comprise multiple modules and regularizations, which is beyond the scope of this paper. We would also like to clarify that proposing new noise type and benchmarking current SOTA methods is not our primary focus. We, instead, focus on the impact of different open-set noise (compared to closed-set noise, and compared with each other) in the standard classification evaluation protocol.
>
> [1] Sachdeva et al. "EvidentialMix: Learning with combined open-set and closed-set noisy labels." WACV. 2021.
>
> [2] Albert et al. "Addressing out-of-distribution label noise in webly-labelled data." WACV. 2022.
>
> >*Q2: Figure 4 (a) and (b) ... it is hard to draw conclusions ...*
>
> A2: Thanks for the comment. Since the noise ratio in Fig. 4(a) and Fig. 4(b) are relatively low, we kindly recommend enlarging the images to better visualize the differences (see an enlarged version in the 1-page PDF for your convenience). We will also update the manuscript. It can be observed that in Fig. 4(a) compared to Fig. 4(b), after a certain epochs of training, the easy open-set noise exhibits significantly different entropy values compared to the clean samples, while the hard open-set noise is difficult to be distinguished. This trend is consistent with what is observed in Fig. 4(c) compared to Fig. 4(d).
>
> >*Q3: Supp E.1 results are confusing...., why the closed-set classification accuracy is always the worst? ... Why is the claim "EntSel + SSR improves open-set detection performance" valid? ...*
>
> A3: We greatly appreciate the thorough review. To clarify, let us briefly revisit the three methods in the appendix: SSR/DivideMix, EntSel, and SSR/DivideMix + EntSel. When integrating EntSel, we adhered to the structure of SSR/DivideMix, which generally comprises two main modules: sample selection and model training. Specifically, we retained the training module of both methods:
> - SSR/DivideMix: the original method.
> - EntSel: we replaced the original sample selection module in SSR/DivideMix with EntSel.
> - SSR/DivideMix + EntSel: we selected the intersection of samples chosen by EntSel and those chosen by the original sample selection module.
> We will revise the manuscript to better clarify the notation.
>
> *Regarding the reasons for the decrease in model performance*, we found that SSR/DivideMix + EntSel selected a significantly smaller subset of samples compared to either SSR/DivideMix or EntSel alone (with default hyperparameters for all). This is expected because the intersection is considered. A possible cause then for the performance decrease is the precision-recall trade-off in the sample selection process. As the reviewer noted, "it selects cleaner samples," which leads to higher precision in sample selection but also results in lower recall. For instance, on the WebVision dataset, this suggests that, for DivideMix/SSR, the negative impact of discarding clean samples to eliminate noisy ones outweighs the negative impact of including slightly more noisy samples. Additionally, the strategy of using intersection is not optimal—effectively integrating open-set sample detection mechanisms with existing sample selection methods presents an interesting area for future research, though it lies beyond the scope of this work.
>
> *"Why is the claim "EntSel + SSR improves open-set detection performance" valid? "* As explained above, we are actually refer to the EntSel method (EntSel) rather than SSR + EntSel. As shown in second row of Fig. 6 and Table 1, we find that replacing the original sample selection module of the SSR method with EntSel may increase the open-set detection performance under open-set noise conditions.
>
> > *Q4: what is the pre-trained weight fitted case used?*
>
> A4: We used a standard `ResNet18` model with pretrained weights from the `torchvision` package. To accommodate the input size, we manually resized the CIFAR-10 dataset to 256x256 in this case. We will update the manuscript to make this clearer.
>
> > *Q5: ... approximation of fitted and memorized classifiers.*
>
> A5: A model $f$ under training is affected by many factors (model capacity / dataset size / training hyperparameters, etc.) - To enable an offline analysis, we would like to note that all theoretical analyses require some approximations/assumptions for tractability. However, the two cases that we introduce, i.e.,: the *memorized case* and the *fitted case*, are realistic and important. The *memorized case* could correspond to overfitting a high capacity model on a small, noisy dataset. By contrast, the *fitted case* could correspond to fine-tuning a linear classifier using a pre-trained model. In Section 4.1, we show experimental results consistent with corresponding theoretical findings respectively.
>
> Thanks once again to the reviewer. If there are any further questions, we would be happy to discuss them at the next stage.

---

> > ### Comment · Reviewer_3Ttr · 2024-08-12
> >
> > I have read the rebuttal. Here are my comments for questions:
> > - Thanks for the new results, this is what I have in mind.
> > - I can see the trend now.
> > - I generally agree with the argument of precision-recall tradeoff, as it was also mentioned in other literatures [1]. It would be interesting to study EvidentialMix + EntSel, as it selects open-set with a  separate interval rather than DivideMix clean/noisy.
> > - Thanks for clarification. Although I do not think resizing is necessary. ImageNet is much larger than CIFAR and CIFAR images is only 32 * 32.
> > - I think this could cause approximation error is not because these two cases are wrong. In general, I agree fitted case and memorized case are realistic. My understanding is that there is no precise way to quantify what is fitted and what is memorized, and of course we can approximate heuristically as what the manuscript does. But it is not fully justified. Neverless, this belongs to grand research direction and is beyond the scope of this paper.
> >
> > My remaining questions are how to benefit downstream tasks with these findings, including integrating with existing methods, effectively improve close/open set classification performance and sample selection.  I am happy to raise my score.
> >
> > [1] Cordeiro, Filipe R., et al. "Longremix: Robust learning with high confidence samples in a noisy label environment." Pattern recognition 133 (2023): 109013.

---

> > > ### Author Response · Authors · 2024-08-12
> > >
> > > Many thanks to the reviewer for the response and recognition. We agree that it would be very interesting to explore how our findings can help improving existing techniques, and this could be a promising direction for the future research. For example, the opposite trends of 'hard' open-set noise and 'easy' open-set noise under the *fitted case* and *memorized case* could potentially suggest useful guidance for sample selection with pre-trained models.

---

### Official Review · Reviewer_6bHU · 2024-07-15

**Soundness:** 3
**Presentation:** 3
**Contribution:** 3
**Rating:** 6
**Confidence:** 2

**Summary:**

The paper refines the problem of learning with noisy labels (LNL) by addressing the often overlooked issue of open-set noise. It provides a comprehensive theoretical analysis comparing the impacts of open-set and closed-set noise, introduces novel datasets for empirical validation, and explores the effectiveness of entropy-based noise detection mechanisms.

**Strengths:**

- The paper offers a thorough theoretical analysis of the differences between open-set and closed-set noise, extending the current understanding of LNL.
- The exploration of entropy-based mechanisms for detecting open-set noise adds a practical tool for improving LNL methods.
- This paper is well-written and easy to understand.

**Weaknesses:**

- The author summarizes two types of open-set noise, i.e., the easy and the hard noise, which is very similar to the symmetric and asymmetric label noise from the perspective of the transition matrix. So does there exist the instance-dependent open-set noise? What is its form if exists?

- In Section 3.5, the author conducts analyses regarding entropy dynamics-based open-set detection, which belongs to the **Fitted case**. If adopting the vision language model (such as CLIP) to fine-tune and detect the open-set noise, is it aligned with the **Memorized case**? It would be better for the author to provide a real-world application for the memorized case.

- The author should clearly illustrate the construct method of closed-set in the experiment (Figure 3) for reproducibility.

**Questions:**

See weaknesses

**Limitations:**

See weaknesses

---

> ### Author Rebuttal · Authors · 2024-08-06
>
> Thanks very much for the careful review. We sincerely appreciate your time and effort in reading our paper, as well as the insightful and positive feedback.
>
> > *Q1: The author summarizes two types of open-set noise, i.e., the easy and the hard noise, which is very similar to the symmetric and asymmetric label noise from the perspective of the transition matrix. So does there exist the instance-dependent open-set noise? What is its form if exists?*
>
> A1: Thanks very much for the question. We would like to note that the proposed *complete noise transition matrix* (L113 in the paper, Definition 3.1) also applies to instance-dependent open-set noise as well. The key difference between instance-dependent noise and the referred symmetric and asymmetric noise (also known as class-dependent noise) is whether we assume that samples from the same class share the same noise transition matrix. Our theoretical analyses are based exclusively on single samples and are therefore applicable to a variety of open-set noise forms.
>
> > *Q2: In Section 3.5, the author conducts analyses regarding entropy dynamics-based open-set detection, which belongs to the Fitted case. If adopting the vision language model (such as CLIP) to fine-tune and detect the open-set noise, is it aligned with the Memorized case? It would be better for the author to provide a real-world application for the memorized case.*
>
> A2: We appreciate the reviewer bringing up such an interesting situation. Theoretically, fine-tuning a large-scale visual language model like CLIP, without freezing the encoder, is likely to cause the model to memorize noisy labels (*memorized case*). We attribute this to the model's tendency to overfit when the dataset predominantly contains a single label for each sample, particularly given the large capacity of deep models (as discussed in lines 166-169 of the paper).
>
> Generally speaking, in most real-world scenarios, when there are enough training budgets (long enough training epochs for example), most of the deep neural networks typically have sufficient capacity to overfit nearly all samples in the dataset. We can thus assume that the *memorized case* corresponds to the learning scenario in most of these situations.
>
>
> > *Q3: The author should clearly illustrate the construct method of closed-set in the experiment (Figure 3) for reproducibility.*
>
> A3: Apologies for the confusion. Since our focus is on open-set noise, we default to the simpler symmetric closed-set noise (randomly flipping the labels of each sample) for more efficient comparison (L 415-416). In the updated manuscript, we will highlight its related details under a separate heading titled "Closed-set Noise.*Though, please note that in our theoretical analysis, we do not make any assumptions about the form of closed-set noise; our theoretical results apply to both symmetric and asymmetric closed-set noise.*
>
> Thanks once again to the reviewer. If there are any further questions, we would be happy to discuss them at the next stage.

---

> > ### Comment · Reviewer_6bHU · 2024-08-11
> >
> > In response to the author's answer to Q3, I have a follow-up question: can the provided theoretical results be extended to instance-dependent noise, where the noise label is strongly correlated with the features?

---

> > > ### Author Response · Authors · 2024-08-11
> > >
> > > Many thanks for the reviewer's reply. We would like to confirm that our theoretical analysis also applies to instance-dependent noise, as we do not assume that samples from the same class follow the same noise transition matrix. We would be happy to engage further disucssions if you have any other concerns.

---

### Author Rebuttal · Authors · 2024-08-06

We thank the reviewers for their insightful comments. We are encouraged that they find our work comprehensive and well-structured, with thorough theoretical analysis and interesting findings. Specifically, we appreciate the recognition of our method's novelty and its empirical validation through novel datasets (**6bHU**, **gSVD**), the significance of the research problem (**6bHU**, **gSVD**, **AwyQ**), the acknowledgment of our method's solid theoretical foundation (**6bHU**, **gSVD**, **AwyQ**) and its interesting/significant findings (**6bHU**, **3Ttr**, **gSVD**).

Here, we would also like to briefly reiterate our key contributions again:
-   Generalized the noise transition matrix to incorporate open-set noise, reformulating the LNL problem.
-   Introduced practical 'memorized' and 'fitted' case scenarios for in-depth offline analysis.
-   Compared open-set and closed-set noise, revealing nuanced impacts on classification accuracy.
-   Differentiated between 'hard' and 'easy' open-set noise, uncovering its contrasting trends in different scenarios.
-   Proposed open-set detection as a complementary evaluation metric and conducted preliminary empirical validations.
-   Created novel synthetic datasets and an open-set test set for rigorous experimentation.

We have provided detailed individual responses to each reviewer's concerns.  If the reviewer has any further questions or concerns, we would be more than happy to engage in additional discussions.

---

### Comment · Area_Chair_Zuh4 · 2024-08-11

Dear Reviewers,

The deadline of reviewer-authors discussion is approaching. If you have not done so already, please check the rebuttal and provide your response at your earliest convenience.

Best wishes,

AC

---

### Decision · Program_Chairs · 2024-09-25

**Decision:**

Reject

**Comment:**

After rebuttal, most reviewers provided borderline and positive rates for this paper due to the solid theoretical analysis and good writing, while there are still several concerns about the presentation and contribution. Specifically, the reviewers suggested that it can be better if the authors can include more discussion about relationship between this work and OOD detection, improve the writing to clarify the contribution to the community. Positive reviewers do not provide strong support for the acceptance. The analysis of open-set noisy labels might be meaningful, but this paper needs to be improved a lot in the writing, which is not ready for publication. So I recommend rejection in this time and encourage the authors to submit to the next conference after improving the presentation.